# Early-life dispersal traits of coastal fishes: an extensive database combining observations and growth models

Marine Di Stefano[1], David Nerini[1], Itziar Alvarez[2], Giandomenico Ardizzone[3], Patrick Astruch[4], Gotzon Basterretxea[2], Aurélie Blanfuné[1], Denis Bonhomme[4], Antonio Calò[5], Ignacio Catalan[2], Carlo Cattano[6], Adrien Cheminée[9,23], Romain Crec'hriou[7], Amalia Cuadros[28], Antonio Di Franco[6], Carlos Diaz-Gil[8], Tristan Estaque[9], Robin Faillettaz[10], Fabiana C. Félix-Hackradt[11], José Antonio Garcia-Charton[12], Paolo Guidetti[24,25], Loïc Guilloux[1], Jean-Georges Harmelin[4], Mireille Harmelin-Vivien[1], Manuel Hidalgo[13], Hilmar Hinz[2], Jean-Olivier Irisson[14], Gabriele La Mesa[15], Laurence Le Diréach[4], Philippe Lenfant[18], Enrique Macpherson[16], Sanja Matić-Skoko[17], Manon Mercader[18], Marco Milazzo[5], Tiffany Monfort[1,9], Joan Moranta[13], Manuel Muntoni[19], Matteo Murenu[20], Lucie Nunez[9], M. Pilar Olivar[21], Jérémy Pastor[22], Ángel Pérez-Ruzafa[12], Serge Planes[26], Nuria Raventos[16], Justine Richaume[9], Elodie Rouanet[4], Erwan Roussel[29], Sandrine Ruitton[1], Ana Sabatés[21], Thierry Thibaut[1], Daniele Ventura[3], Laurent Vigliola[27], Dario Vrdoljak[17], and Vincent Rossi[1]

[1]Mediterranean Institute of Oceanography (MIO, UM 110, UMR 7294), Aix Marseille Univ., Univ. Toulon, CNRS, IRD, Marseille, 13288, France

[2]Mediterranean Institute for Advanced Studies, IMEDEA (UIB-CSIC), Miquel Marques 21, 07190 Esporles, Balearic Islands, Spain

[3]Marine Ecology and Biology Lab, Department of Environmental Biology, Sapienza University of Rome, Rome, Italy

[4]GIS Posidonie, Case 901, Campus de Luminy, 13288, Marseille Cedex 9, France

[5]Department of Earth and Marine sciences (DiSTeM), University of Palermo, Via Archirafi 20-22, 90123, Palermo, Italy

[6]Stazione Zoologica Anton Dohrn, Department of Integrative Marine Ecology, Sicily Marine Center, Lungomare Cristoforo Colombo (complesso Roosevelt), 90149, Palermo, Italy

[7]Station Biologique CNRS, Sorbonne Université, Service Observation, Place Georges Teissier CS90074, 29688, Roscoff, France

[8]Xelect Ltd., St. Andrews KY16 9LB, UK

[9]Septentrion Environnement, 89 Traverse Parangon, 13008, Marseille, France

[10]DECOD (Ecosystem Dynamics and Sustainability), IFREMER, INRAE, Institut Agro, Lorient, France

[11]Marine Ecology and Conservation Lab., Centre of Environmental Science, Universidade Federal do Sul da Bahia, Rod Joel Maers BR 367, km 10, CEP: 45810-000 Porto Seguro, Bahia, Brazil

[12]Departamento de Ecología e Hidrología, Universidad de Murcia, Campus Mare Nostrum de Excelencia Internacional, 30100, Murcia, Spain

[13]Spanish Institute of Oceanography (IEO, CSIC), Balearic Oceanographic Center (COB), Ecosystem Oceanography Group (GRECO), Moll de Ponent s/n, 07015 Palma, Spain

[14]Sorbonne Université, CNRS, Laboratoire d'Océanographie de Villefranche (LOV), Villefranche-sur-Mer, France

[15]Italian Institute for Environmental Protection and Research (ISPRA), Roma, Italy

[16]Centre d'Estudis Avançats de Blanes (LEOV-CEAB-CSIC), Otolith Research Lab, Car. Acc. Cala St. Francesc 14, 17300, Blanes, Girona, Spain

[17]Institute of Oceanography and Fisheries, Šetalište Ivana Meštrovića 63, 21000 Split, Croatia

[18]Université de Perpignan Via Domitia, Centre de Formation et de Recherche sur les Environnements Méditerranéens, UMR 5110, 52 Avenue Paul Alduy, F-66860, Perpignan, France

[19]Groupement d'Intérêt Public Seine Aval, Hangar C, Espace des Marégraphes, Quai de Boisguilbert, 76000 Rouen

[20]Department of Life and Environmental Science, University of Cagliari, Italy

[21]Institut de Ciències del Mar (ICM-CSIC), Passeig Marítim de la Barceloneta, 37-49, Barcelona 08003, Spain

[22]48 rue Rouget de l'Isle, 66310, Estagel, France

[23]Faculté des Sciences, Aix Marseille Université, 163 Avenue de Luminy, Case 901, 13288 Marseille, France

[24]Department of Integrative Marine Ecology (EMI), Stazione Zoologica Anton Dohrn, National Institute of Marine Biology, Ecology and Biotechnology, Genoa Marine Centre, Villa del Principe, Piazza del Principe 4, 16126 Genoa, Italy

[25]National Research Council, Institute for the Study of Anthropic Impact and Sustainability in the Marine Environment (CNR-IAS), Via de Marini 6, 16149 Genoa, Italy

[26]PSL Research University, EPHE-UPVD-CNRS, UAR 3278 CRIOBE, Université de Perpignan, 66860 Perpignan Cedex, France

[27]UMR ENTROPIE, IRD-UR-UNC-IFREMER-CNRS, Centre IRD de Nouméa, Nouméa Cedex, New-Caledonia, France

[28]Instituto Español de Oceanografía, Centro Oceanográfico de Cádiz, Puerto Pesquero, Muelle de Levante, s/n, PO Box 2609, E-11006, Cádiz, Spain

[29]Independent researcher

**Correspondence:** Marine Di Stefano (marinedistefano18@gmail.com) and Vincent Rossi (vincent.rossi@mio.osupytheas.fr)

**Abstract.** Early-life stages play a key role in the dynamics of bipartite life cycle marine fish populations. Difficult to monitor, observations of these stages are often scattered in space and time. While Mediterranean coastlines have been highly surveyed, no effort was made to assemble historical observations. Here we build an exhaustive compilation of dispersal traits for coastal fish species, considering in-situ observations and growth models.

Our database contains over 110 000 entries collected from 1993 to 2021 in various subregions. All observations are harmonized to inform on dates and geolocations of both spawning and settlement, along with pelagic larval durations. When applicable, missing ~~dates~~ data and associated confidence intervals are reconstructed from Dynamic Energy Budget theory.

Statistical analyses allow revisiting traits' variability and revealing sampling biases across taxa, space and time, hence providing recommendations for future studies and sampling. Comparison of observed and modelled entries gives hints to improve the feed of observations into models. Overall, this ~~long-term~~ extensive database is a crucial step to investigate how marine fish populations respond to global changes across environmental gradients.

## 1 Introduction

Pelagic early-life stages of marine organisms with bipartite life cycle are key to understand spatiotemporal dynamics of marine populations (Dubois et al., 2016), especially for fish species with high site fidelity as adults. These stages have pronounced dispersive abilities and are highly sensitive to environmental factors, leading to potentially high mortality rates, ultimately influencing juveniles' replenishment (Gaines et al., 2007; Hidalgo et al., 2019). For benthic or demersal coastal fish, early-life is decomposed into different stages for which many different classifications and names exist due to their complexity. They are generally based on ontogenic characters, age and size and sometimes differ from one species to another, as summarized by Vigliola and Harmelin-Vivien (2001). Building upon comprehensive classifications of Kendall et al. (1984) and Catalan et al. (2014), we here use a simplified life cycle based on sizes and generalized wording as it follows. Gametes are first released during spawning events and eggs are fertilized almost immediately, marking the start of the dispersive phase; they

then hatch into larvae drifting and eventually swimming toward coastal areas; individuals become settlers as soon as they settle in nurseries, considered here as the end of the early-life dispersive phase. After settlement, juvenile fish grow in nurseries generally until maturity, when they recruit into adult populations (these parts of the life cycle are not considered here).

Studying these early-life stages requires a good knowledge of when and where propagules (eggs and larvae) are released in the water column during spawning (Di Stefano et al., 2022), of their dispersal routes (Legrand et al., 2019) and of conditions faced during this dispersive phase (Torrado et al., 2021). Indeed, as small organisms, development, growth and survival of eggs and larvae are highly impacted by abiotic factors such as changes in temperature or oxygen, but also by other biotic pressures such as predation or food availability (Pineda et al., 2007). Ocean currents also influence dispersal trajectories, affecting

potential preys and conditions encountered along their drift, as well as the settlement success in nurseries (White et al., 2019). Thus, population connectivity of bipartite life cycle fish species is mainly driven by early-life stages, especially for relatively sedentary species (Gaines et al., 2007). However, the difficulty in monitoring these stages *in-situ* (costly, time-consuming) leads to a paucity of observations, scattered in space and time. It thus limits our understanding of the control they exert on population dynamics, especially over long timescales and when considering the tremendous variability of oceanographic processes (Bates

et al., 2018).

     Nevertheless, many coastlines, such as in the Mediterranean Sea, have been highly studied ~~in last~~ over several decades with ~~several~~ independent research projects carried out in various sub-regions (Harmelin-Vivien et al., 1995; Olivar et al., 2014). Exploited for centuries (Tsikliras et al., 2015) and recognized as a climate-change hotspot (Lionello & Scarascia, 2018; Soto-Navarro et al., 2020), the Mediterranean Sea and its coastlines are biodiversity hubs (Myers et al., 2000; Coll et al., 2010)

and are well suited to address the role of anthropogenic pressures and climate change on marine populations (Lejeusne et al., 2010; Matić-Skoko et al., 2020). For instance, Bianchi and Morri (2003) report increasing occurrences of warm-water species and Marbà et al. (2015) review the warming-induced impacts on the survival and phenology of Mediterranean biota. However, ecological studies with long-term and broad-scale perspectives are limited, especially because observations are rarely pooled together, even when performed with similar sampling methods (e.g. Faillettaz et al., 2020).

Inspired by review studies on fish biological traits, such as spawning periods based on gonadosomatic index (Tsikliras et al., 2010) or Pelagic Larval Durations (PLD) based on otolithometry (Macpherson & Raventos, 2006), as well as comparative studies of different sampling methods (e.g. Catalan et al., 2014), our goal here is to build an extensive compilation of historical data of early-life traits from multiple sources. We focus on the Mediterranean basin to benefit from relatively high data availability while addressing both knowledge gaps on fish phenological traits (Daskalaki et al., 2022) and on early-life stages

connectivity (Hidalgo et al., 2017). As such, we aim ~~at gathering~~ to gather the more numerous published and unpublished data on Mediterranean marine coastal fish species (benthic and demersal) to provide the most comprehensive information to-date on their traits. To do so, one original~~ity~~ aspect of this work is to use growth models from the Dynamic Energy Budget (DEB) theory (Marques et al., 2018; Kooijman et al., 2020) ~~allowing~~ which allows us to enrich observations with modelled estimates in a coherent and robust manner. Such a database would help to provide evidence and interpret ongoing changes of fish population

dynamics, linked to climate change and anthropogenic threats occurring in the Mediterranean Sea.

In this study, we pool data collected over ~~the past~~ several decades with various standardized methods in multiple studies (including both direct observations and indirect methods or models), and harmonize them into a uniform dataset of dates and locations of both spawning and settlement, as well as of PLDs. When possible, missing information has been reconstructed based on the online Add-My-Pet database (AmP; Kooijman, 2022). In its present state, the database also encompasses several taxa, including some patrimonial and exploited coastal fish species, and provides measures of uncertainties and sampling characteristics. Statistical analyses of this consolidated database allow ~~describing~~ us to describe overall taxonomic and spatiotemporal ~~covers~~ coverage and ~~evaluating~~ evaluate potential sampling gaps. We finally discuss our original approach and suggest future research directions for both modelling and observational perspectives, including connectivity or trait-ecology studies at climatic and basin scales that this database should promote.

## 2  Methods

### 2.1  Database construction

#### 2.1.1  Compilation procedure

The screening of studies focusing on early-life stages of coastal fish species was conducted in November 2021. A PRISMA flow diagram summarizes all different steps (see Figure S1). For this compilation, we gather 44 datasets sampled in the North-Western and Central Mediterranean Sea (including Adriatic Sea), shared by 51 data providers (see Authors of the article and Table S1). Data are ~~sampled~~ collected along Spanish, French, Italian, Croatian and Montenegrin coastlines. The database covers a period from 1993 to 2021, spanning 29 years of observations. Entries implemented in this database are obtained by diverse standardized sampling methods (direct and indirect measures of early-life traits), depending on studies' aim: otoliths' data (extracted from individuals collected by hand nets), datasets of settlers and juveniles sampled by Underwater Visual Census (UVC), shore seine, light-traps, as well as fish eggs and larvae sampled with plankton nets. Each of these techniques are standardized, peer-reviewed and widely accepted by the scientific community (see "Material and Methods" of the cited papers). The actual database represents entries characterized by sampling information, taxa, early-life dates and locations. Each row of the database is called an entry and may concern multiple individuals at a specific stage (i.e. eggs, larvae, settlers or juveniles). Each column is called a variable.

#### 2.1.2  Harmonisation procedure

Spawning date is defined as the exact day when eggs are released by adults in open water (Kendall et al., 1984). As some sampling methods sampled just-hatched larvae, their spatiotemporal information is also considered as linked to spawning when length is not available. Review of Pauly and Pullin (1988) records a mean of 2 days and a maximum of 5 days between spawning and hatching for Mediterranean fish species, which is acceptable given the range of uncertainty taken into account in the rest of the study (see section 2.2). Settlement date is defined as the transition between larval pelagic stage and juvenile coastal stage (Kendall et al., 1984; Vigliola & Harmelin-Vivien, 2001) and is known to occur at a rather stable length for each

Mediterranean coastal fish species (Raventos et al., 2021). The PLD is defined as the time of pelagic dispersal between these two events (spawning and settlement).

Some sampling techniques directly provide the required information on both events whereas other methods required data processing (Figure 1). In fact, samples gathered in this database are not necessarily caught during studied phenological events but sometimes right after (methodology described in section 2.2). For instance, otolithometry provide information on both spawning and settlement dates (Di Franco & Guidetti, 2011; Cattano et al., 2017), whereas UVC method only provides the sampling date of juveniles (Cuadros et al., 2017; Mercader et al., 2019). In this latter case, both spawning and settlement dates have been determined ~~thanks to~~ using a DEB model-based approach (see section 2.2).

Dates of spawning and settlement are recorded as `SpawningDate_mean` and `SettlementDate_mean` variables (see section 2.2 for further information on averages). `SpawningDetermination` and `SettlementDetermination` variables explain how these dates are determined (reconstruction, sampling date or otolithometry). The Pelagic Larval Duration (`PLD_mean` variable) is computed as the number of days between these two dates (Green et al., 2009). Along with the mean phenological information, the standard deviation of each entry is stated as `SpawningDate_std`, `SettlementDate_std` and `PLD_std` variables, in days (see section 2.2 for further information on standard deviation).

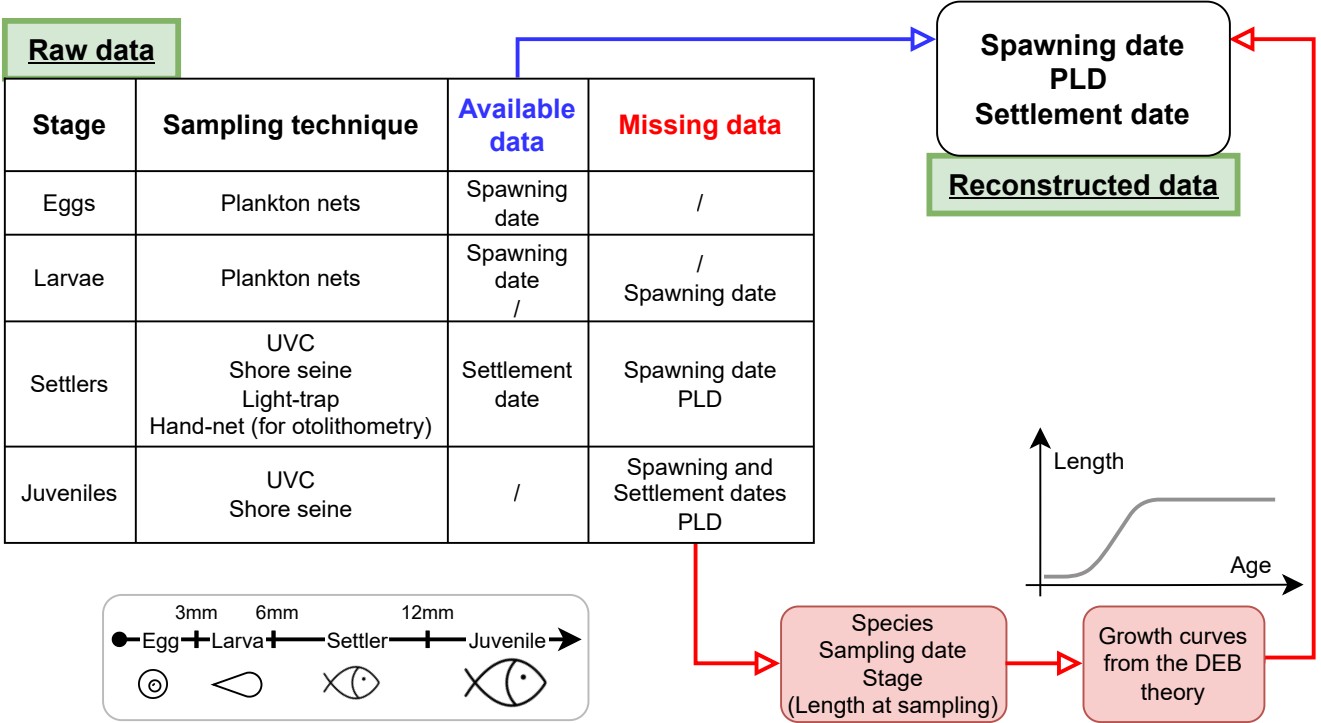

**Figure 1.** Workflow applied to each entry of the database, as a function of stage, sampling technique and available information (e.g. date of spawning or settlement, available or missing) to obtain the database. Length category associated to each stage is also displayed.

Spatial information is recorded in the database as `Latitude` and `Longitude` variables, in decimal degrees. Thus, these coordinates are associated to a specific event, spawning or settlement, in the `CoordinatesType` variable, depending on the stage of the individual and on the sampling method. For methods giving information on eggs and just-hatched larvae (e.g. bongo net), the sampling location has been used as a proxy of the spawning location. For juvenile datasets, we assume that juveniles live in the same area as settlers in nurseries, so the sampling location relates to the settlement location.

Other general sampling variables are implemented if available (see Table S2 and metadata of the database in SEANOE). The main information used in following sections is the `SamplingDate` variable along with the `SamplingLength` variable when available, standing for the total length of individuals in millimetres, but also the `Stage` variable and necessarily, the taxon (`Family`, `Genus` and `Species` variables). However, as the sampling design changes between entries, specific sampling information might be sometimes missing for some entries and replaced by NA (Not Available).

## 2.2   Reconstruction of missing data

One ~~originality of this work~~ unique feature of this database is the compilation and harmonization of data collected with different sampling methods. Some allow direct access to dates of spawning and/or settlement. Others do not allow the timing of both events to be estimated. A process commonly used to indirectly estimate these missing dates based on species-dependent growth models (or age-length curves) is back-calculation modelling. In this study, species-specific growth models based on Dynamic Energy Budget theory (DEB; Kooijman et al., 2020; Kooijman, 2022) are used to estimate missing spawning and/or settlement dates. These growth models are freely available on the Add-my-Pet database (AmP; already used to study fish, see van der Meer and Kooijman (2014)) which compiles metabolic and energetic information on a large number of animal species.

### 2.2.1   DEB theory and models

DEB theory applies for the study of Age-Length growth of a large variety of organisms as it uses models describing metabolic processes by exploiting conservation law for energy and mass (Agüera et al., 2017). It is thus adaptable to any homeostatic system exchanging flows of energy and mass between inside and outside, at the individual level (Kooijman, 2010). DEB theory is not only based on data to build models, but also on the general shape of species and on the resultant information on exchanges of the organism with environment. Thus, it makes this theory highly stable regardless of input data, which ~~comforts~~ validates and substantiates our choice of DEB models to describe individual-specific growth (Marques et al., 2018). These models also integrate the effect of environment on organisms by considering food intake or ambient temperature when evaluating model parameters (Kooijman, 2010). It helps ~~determining a more realistic physiology of organisms and their~~ to provide a realistic physiological basis of growth dynamics modulated by environmental variability all along their life cycle (Marques et al., 2018).

In this theory, different growth models are described according to different life stages. Their classification is based on the type of metabolic acceleration, to better fit with species life cycle. The common DEB growth model denoted as `std` model is based on Von Bertalanffy growth curve (Table 1). It is supposed to apply to species without prominent larval phases and morphological metamorphosis. AmP database proposes this model for only a few fish species (*Chelon labrosus*, *Chelon*

*ramada*, *Chelon saliens*, *Gobius niger*, *Gobius paganellus*, *Macroramphosus scolopax*, *Scomberesox saurus*, *Zeus faber* and also *Mugil liza*, *Mugil cephalus*, *Mugil curema* species used for the genus-based reconstruction of *Mugil sp.*). The DEB growth model used for all other species is adapted to a bipartite life cycle (Table 1; Kooijman, 2010). Denoted as `abj` model, the latter splits the growth curve in two parts. The first part describes exponential growth of the larval phase (metabolic acceleration) finishing at morphological metamorphosis. The second part is a Von Bertalanffy growth curve.

While the growth models advised by the DEB theory (`std` or `abj`) are used here, it is worth noting that the original Von Bertalanffy growth model (basis of the `std` model, excluding $t_j$; Von Bertalanffy, 1938) has been largely questioned in fisheries science as it obviously lacks biological realism in excluding early-life stages (Vasbinder & Ainsworth, 2020). The `abj` model is indeed built upon a modified Von Bertalanffy growth curve (including $t_j$; Beverton and Holt, 1957), while adding an exponential curve to best describe larval growth (Vasbinder & Ainsworth, 2020).

**Table 1.** ~~Two types of g~~ Growth models used for reconstruction of missing PLD, spawning and settlement dates. ~~Parameters are length $L$, length at birth $L_b$, length at metamorphosis $L_j$, asymptotic length $L_i$, age $t$, age at metamorphosis $t_j$, specific growth rate during acceleration $r_j$ and Von Bertalanffy growth rate $r_B$ (Kooijman, 2010).~~ Length~~s are~~ is in millimetre $mm$, age~~s~~ in day $d$ and growth rate~~s~~ in $d^{-1}$ (Kooijman, 2010). Except $L_b$ directly extracted from AmP, all parameter values are susceptible to change with food and temperature conditions.

| | Model `std` (Von Bertalanffy) | Model `abj` (exponential & Von Bertalanffy) |
|---|---|---|
| **Equations** | $L = L_i - (L_i - L_b)\exp(-r_B t)$ | If $t < t_j$: $\qquad L = L_b \exp(\frac{r_j t}{3})$ <br> If $t \geq t_j$: $\qquad L = L_i - (L_i - L_j)\exp(-r_B(t - t_j))$ |
| **State variables** | $L$: length <br> $t$: age | $L$: length <br> $t$: age |
| **Parameters** | $L_b$: length at birth <br> $L_i$: asymptotic length <br> $r_B$: Von Bertalanffy growth rate | $L_b$: length at birth <br> $L_i$: asymptotic length <br> $r_B$: Von Bertalanffy growth rate <br> $L_j$: length at metamorphosis <br> $t_j$: age at metamorphosis <br> $r_j$: specific growth rate during acceleration |

### 2.2.2   Reconstruction process

The reconstruction process can concern PLD, spawning date $D_b$ and settlement date $D_{settl}$. Four different stages are considered in the organism early-life time. Their specific length ranges are adapted from the generalized life cycle of littoral demersal Mediterranean fish species suggested by Catalan et al. (2014): 1) eggs and just-hatched larvae ($< 3$ mm); 2) larvae (3 - 6 mm); 3) settlers (6 - 12 mm); 4) juveniles ($> 12$ mm). These size ranges determine which trait can be estimated (see Figure S2). Note that, strictly speaking, hatch differs from birth. Hatch is the event when larvae free themselves from the egg membrane; birth generally refers to the time when larvae start feeding, as in some cases mouth needs slightly more time to open. During this short time interval (spanning a few hours to a few days for slow-development species), development relies on the yolk sac,

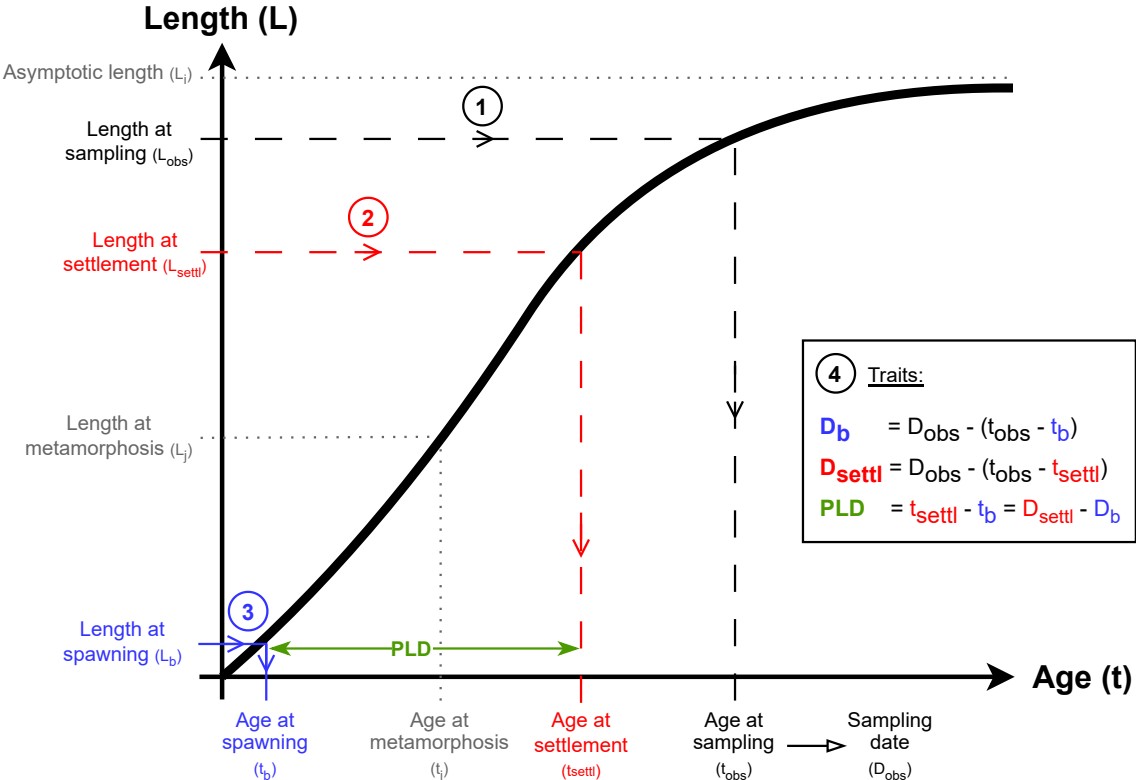

**Figure 2.** Summary of the reconstruction process for one exemplary species-specific growth curve (black thick line). Except $L_b$ directly extracted from AmP, all parameters are food ($f$) and temperature ($T$) dependent. Circled numbers represent the successive steps of the reconstruction. Blue is associated to spawning, red to settlement, green to PLD and black to sampling. Grey annotations and parameters ($L_j$, $L_i$, $t_j$) do not intervene in the reconstruction (see also Table 1). ~~Step-by-step summary of the reconstruction process for an entry of a fish species whose growth curve (black thick curve) is given by Add-My-Pet. The number represents the different steps of the reconstruction. Blue is associated to spawning, red to settlement, green to PLD and black to sampling. Some informative parameters ($t_j$, $L_j$ and $L_i$) used in equations of Table 1 are displayed in grey.~~

whose energy is a function of the mother's investment into reproduction (Kooijman, 2010). Here, we disregard the distinction between hatch and birth and the associated duration by assuming that hatching is the beginning of fish early-life stages.

Consider an Age-Length growth curve of a given fish species with known parameters (Table 1 and black curve on Figure 2). Consider now a length at sampling $L_{obs}$ of that fish species at sampling date $D_{obs}$, in $L_{settl} < L_{obs} < L_i$ so that spawning date $D_b$, settlement date $D_{settl}$ and $PLD$ can be estimated. Using the reciprocal growth model with entry $L_{obs}$, we get the age at sampling $t_{obs}$ (step 1 in Figure 2). In the same way, the age at settlement $t_{settl}$ and the age at spawning $t_b$ are deduced respectively from the length at settlement $L_{settl}$ (step 2 in Figure 2) and the length at spawning $L_b$ (step 3 in Figure 2). Finally,

PLD, spawning date $D_b$ and settlement date $D_{settl}$ are estimated using equations in Figure 2 (step 4, $PLD = t_{settl} - t_b$; $D_b = D_{obs} - (t_{obs} - t_b)$ and $D_{settl} = D_{obs} - (t_{obs} - t_{settl})$). When $L_{obs} \geq L_i$, the estimation process is not reliable and PLD, spawning and settlement dates are set to NA values in the database.

Sources of variability in previous estimates are twofold. First, regardless of the DEB model used in Table 1, parameters that control growth curve shape are indeed dependent on environmental factors encountered by organisms throughout their lifetime. As specified by Kooijman (2022) and Kooijman (2010), DEB growth model parameters vary with temperature ($T$) and food ingestion capability ($f$). Details for computation of length and growth parameters are given in AmP codes (Kooijman, 2022; Kooijman, 2010). Finally, length at settlement $L_{settl}$ is the second source of variability since its value changes between individuals.

It is possible to include these different sources of variability in the estimation process when taking some values for $T$, $f$ and $L_{settl}$ ranging in ~~reasonable~~ realistic intervals. It is admitted that annual variations of Mediterranean seawater temperature range from 15 to 25 °C (García-Monteiro et al., 2022)). According to Kooijman (2010), food ingestion capability $f$ is a quantity lying between 0 (absence of food) and 1 (food *ad libitum*). A rather favourable range of $f \in [0.7, 1]$ ~~is~~ was selected arbitrarily ~~chosen~~, since ~~sampled~~ collected individuals have survived their early-life stages. Indeed, lower values of food would have reflected that individuals could have died during dispersal. Finally, Catalan et al. (2014) suggest that length at settlement ranges from 6 to 12 mm for most littoral demersal Mediterranean fish species. However, some genus like *Atherina sp.*, *Lipophrys sp.* and *Hippocampus sp.* present length at birth $L_b$ (= length at hatching) higher than 6 mm in AmP parameters. In this case, choice is made to randomly pick length at settlement $L_{settl}$ between 12 to 30 mm.

For each entry, the reconstruction process described in Figure 2 is reiterated 50 times (see Figure S6) using values of $T$, $f$ and $L_{settl}$ uniformly drawn from their respective range. Estimations of PLD, spawning date $D_b$ and settlement date $D_{settl}$ are then taken as the mean value of these 50 estimations (Table 2). Empirical standard deviations (Table 2) are also computed (recorded as `SpawningDate_mean` and `SpawningDate_std`, `PLD_mean` and `PLD_std`, `SettlementDate_mean` and `SettlementDate_std` in the database; see Figure S7 for the range of PLD variability).

**Table 2.** Equations of means and associated standard deviations of PLD, spawning date $D_b$ and settlement date $D_{settl}$ for $N = 50$ reconstructions where environmental conditions are randomly drawn from $T \in [15, 25]$, $f \in [0.7, 1]$ and $L_{settl} \in [6, 12]$ (or $L_{settl} \in [12, 30]$).

| Mean | Standard deviation |
|---|---|
| $\overline{PLD} = \frac{1}{N} \sum_{i=1}^{N} PLD_i$ | $\sigma_{PLD}^2 = \frac{1}{N} \sum_{i=1}^{N} (PLD_i - \overline{PLD})^2$ |
| $\overline{D}_b = \frac{1}{N} \sum_{i=1}^{N} D_{b_i}$ | $\sigma_{D_b}^2 = \frac{1}{N} \sum_{i=1}^{N} (D_{b_i} - \overline{D}_b)^2$ |
| $\overline{D}_{settl} = \frac{1}{N} \sum_{i=1}^{N} D_{settl_i}$ | $\sigma_{D_{settl}}^2 = \frac{1}{N} \sum_{i=1}^{N} (D_{settl_i} - \overline{D}_{settl})^2$ |

~~55~~ Fifty-five fish species are reconstructed based on the list of species available in AmP (see Table S4), using the parameters extracted from the AmP portal on the 11 July 2022 (https://www.bio.vu.nl/thb/deb/deblab/add_my_pet/). In `SpawningDe-termination` and `SettlementDetermination` variables, these entries are defined by the "reconstruction species-

based" category. If a species is not in AmP (110 species) or only represented by a genus, a mean growth curve is constructed using available AmP species curves from the same genus (see Tables S4, S5 and S6). Reconstruction is thus defined as "genus-based" in `SpawningDetermination` and `SettlementDetermination` variables (see Tables S5 and S6). In other cases, NA is attributed to all traits in the database variables.

## 2.3 Statistical analyses

In Results section, temporal distributions of sampling, spawning and settlement periods are displayed as circular-plots using Von Mises kernel density estimates (Fisher, 1993). These representations allow observing seasonality of traits, considering the periodicity of the studied processes. Data in "sampling date" or "otolithometry" subcategories of `SpawningDetermination` and `SettlementDetermination` variables are considered observed data. Species-based or genus-based reconstructed dates are considered reconstructed data. Circular uniform probability distributions are displayed as a baseline to simulate uniform probabilities over an annual cycle.

In section 3.2, PLD data are grouped into similar categories (e.g. observed, reconstructed and composite) whose distributions are fitted using Gaussian mixture models (Bouveyron et al., 2019). Observed PLDs come from otolithometry data since spawning and settlement dates are fully derived from observations. Reconstructed PLDs are represented by juvenile data as both spawning and settlement dates are reconstructed. Finally, composite PLDs are estimated when mixing observed settlement date and reconstruct spawning date (settlers data).

## 3 Results

### 3.1 General description of the database

The total number of entries (rows in the database) is ~~118 666~~ 118 661, among which 79 645 entries (67.1% of the database) inform on both spawning and settlement, thus allowing the computation of PLDs. Dates and locations of spawning concern ~~95 801~~ 95 796 entries (80.7%), while the same information on settlement concerns 82 801 entries (69.8%). Some entries only characterize spawning (~~16 156~~ 16 151 rows, 13.6%) or settlement (3 156 entries, 2.7%), thus preventing the estimation of PLDs. Overall, 19 709 entries (16.6%) provide only information on spawning or settlement locations but not on dates.

A total of ~~56~~ 55 families, 102 genera and 165 species are recorded in the database (see Table S3). The most represented family is Sparidae with 38 259 entries (32.2%), the genus is *Diplodus* with 20 416 entries (17.2%) and the species is *Chromis chromis* with 12 914 entries (10.9%) (Figure 3). Note that there are more entries when enumerated per family and per genus than per species due to the proportion of data for which only the family (~~9 863~~ 9 858 entries, 8.3% of the database) or the genus (5 110 entries, 4.3% of the database) are specified (e.g. eggs and larvae sampled by plankton nets are difficult to identify at species-level, as well as very young juveniles sampled by ~~Underwater Visual Census,~~ UVC).

This database combines information derived from 11 general sampling techniques. Entries ~~are~~ mainly ~~coming~~ come from UVC (72 860 entries, see Figure S3a) followed by shore seines (20 555 entries), both of which capturing essentially juvenile

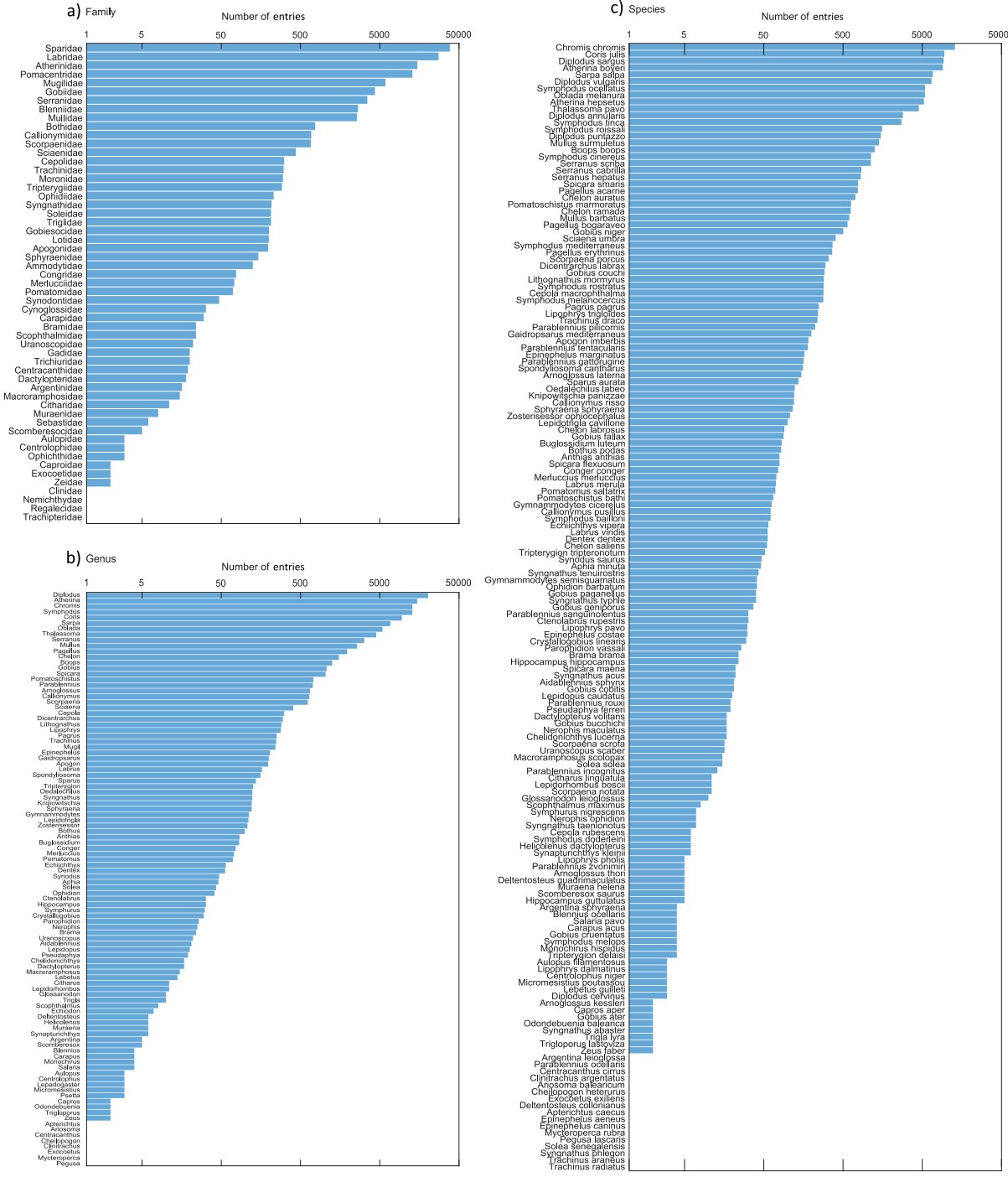

**Figure 3.** Number of entries (logarithmic scale) per a) family, b) genus and c) species.

fishes (89 458 entries, see Figure S3b). They have been generally collected near the surface: 90% of the data are associated to depths ranging from 0 to 10 m, while the maximum depth recorded in the database is 175 m, sampled with a plankton net.

Concerning the spatial coverage of the database, the most represented ecoregion as defined by Spalding et al. (2007) is the Western Mediterranean with ~~96 460~~ 96 455 entries, followed by the Adriatic Sea with 21 764 entries, comparatively to the least represented regional one, the Alboran Sea with 434 entries and the Ionian Sea with only 8 entries. Note that the sampling has been spatially-widespread in some regions while concerning a low numbers of entries (Catalan shorelines for instance), whereas some locations (very restricted in space, such as reference sites or institutional time-series) have been repeatedly sampled over time to study temporal variability (Figure 4). Moreover, certain areas are represented by multiple species (for instance the Balearic archipelago, concerned by generalist sampling techniques such as bongo nets) whereas some are limited to one to a few species (Italian coasts in general, focusing especially on a few Sparidae species such as *Diplodus sargus*). Overall, areas with the highest number of entries are coastlines from Marseilles to Nice, the border between France and Spain, the Balearic archipelago (especially in Menorca and in the South-West of Majorca Island), coastlines around Murcia and Croatian shores.

Considering time dimension, the database covers 29 years (1993 to 2021) with a noticeable gap of sampling efforts around the beginning of the 2000s. The 2010s are well sampled while the most recent years (around 2020s) are not (Figure 5). Over an annual cycle, one can see that sampling has been mainly performed from May to early August and in September-October (Figure 6a). There is a clear lack of sampling in winter from December to mid-March but also in early spring (late-March

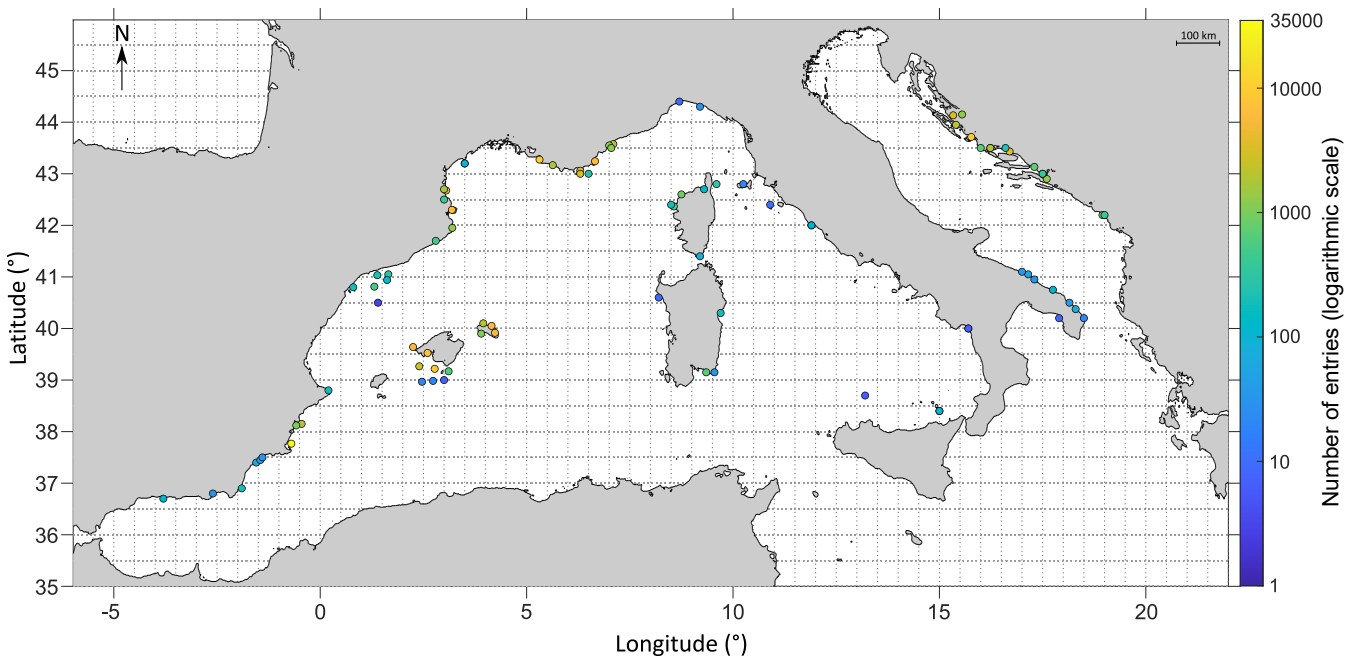

**Figure 4.** Number of entries sampled per node. Each node is 0.5° width per 0.5° long, which represents around 50 km width per 50 km long. The coordinates system used is WGS84.

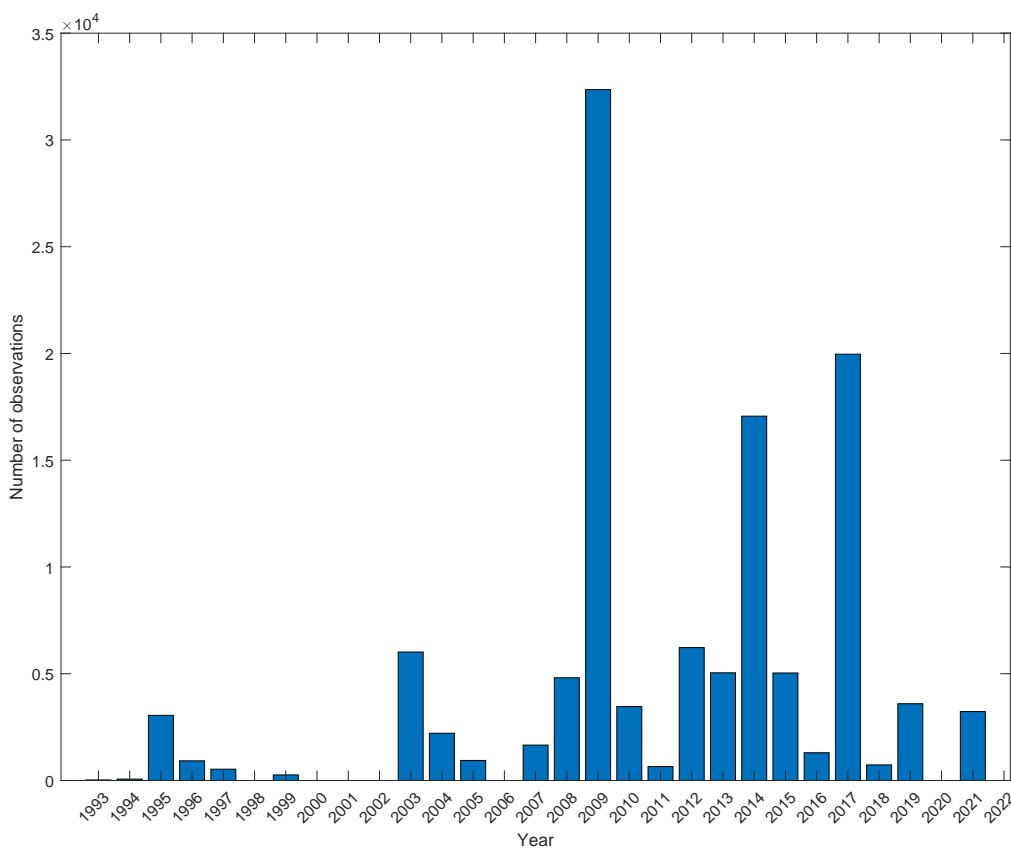

**Figure 5.** Number of entries sampled per year from 1993 to 2021.

to April). All spawning and settlement dates cover the whole year (bold black curves on Figure 6b and c). More precisely, spawning prevails from March to mid-June and from July to mid-August, while weakening from September to January (Figure 6b). Regarding settlement, it spans February to June, with local minima from July to September and from October to January (Figure 6c, black curve). Across the entire database, PLDs range from a minimum of 4 days to a maximum of 250 days. Density estimates of PLDs reveal three distinct peaks at 20 days, 37 days and 64 days (Figure 7, bold black curve).

### 3.2 Comparison between observed and reconstructed data

To evaluate the impact of combining direct and indirect estimates, we grouped data into two broad categories: observed data (i.e. determined directly at sampling or thanks to otolithometry) versus reconstructed data (i.e. reconstructed using DEB theory, based on species or genus information). Observed data represent ~~15 102~~ 15 097 entries for spawning dates (15.8% of total spawning data) and 14 343 entries for settlement dates (17.3% of total settlement data). Reconstructed dates sum up to 80 699

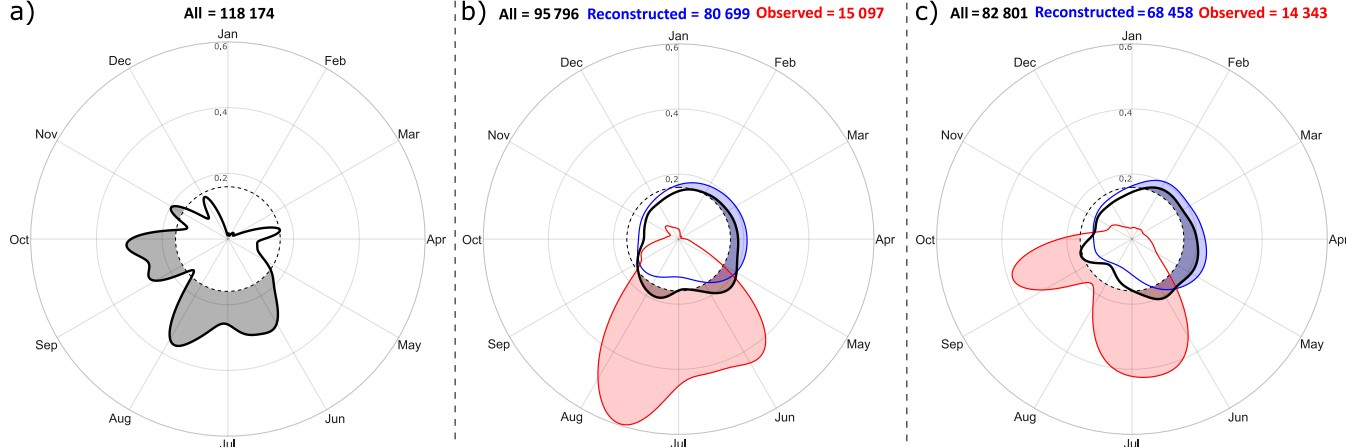

**Figure 6.** Kernel circular-density estimates of the a) sampling, b) spawning and c) settlement comparing all data (bold black curves), observed data (red curves) and reconstructed data (blue curves). Shaded area under a contour line displays high density regions above uniform distribution (dashed circle). No sampling date is available for some otolithometry data (487 entries), thus number of sampling dates in a) is not equal to number of entries in the database.

entries for spawning (84.2% of total spawning data) and 68 458 entries for settlement (82.7% of total settlement data), from which species-based reconstruction represents respectively 64 154 spawning entries (79.5% of spawning reconstruction and 67.0% of total spawning data) and 55 847 settlement entries (81.6% of settlement reconstruction and 67.4% of total settlement data). Consequently, remaining 16 545 spawning entries (20.5% of reconstructed spawning dates and 17.3% of total spawning data) and 12 611 settlement entries (18.4% of reconstructed settlement dates and 15.2% of total settlement data) concern genus-based reconstructions.

Observed spawning dates occur from mid-May to mid-August (Figure 6b, red curve); observed settlement dates mainly take place from mid-May to late-July and from mid-August to late-September (Figure 6c, red curve). While this temporal shift of a few months could be partly explained by the expected delay between spawning and settlement events, there is also an evident lack of information for spawning from September to April and for settlement from October to May as well as in August. Reconstructed dates for both spawning and settlement are rather well distributed over an annual cycle (Figure 6b and 6c, blue curves), although slightly less information is available from June to late-December.

For both spawning and settlement, annual distributions of all data are very similar to those of reconstructed data, being less skewed than observed data. The overall shape of all data is however influenced by observed ones, as shown by the seasonal maxima from May to August for spawning and from June to late-September for settlement. Nevertheless, annual distributions of reconstructed and observed data are quite different, as are all and observed data. In fact, spawning (settlement respectively) mainly occurs at distinct periods for each of these categories.

PLDs are separated into three broad categories: fully observed data represent 2 181 entries (2.7% of total PLD data), fully reconstructed data represent 68 458 entries (86.0% of total PLD data) and composite data (e.g. PLDs estimated from observed

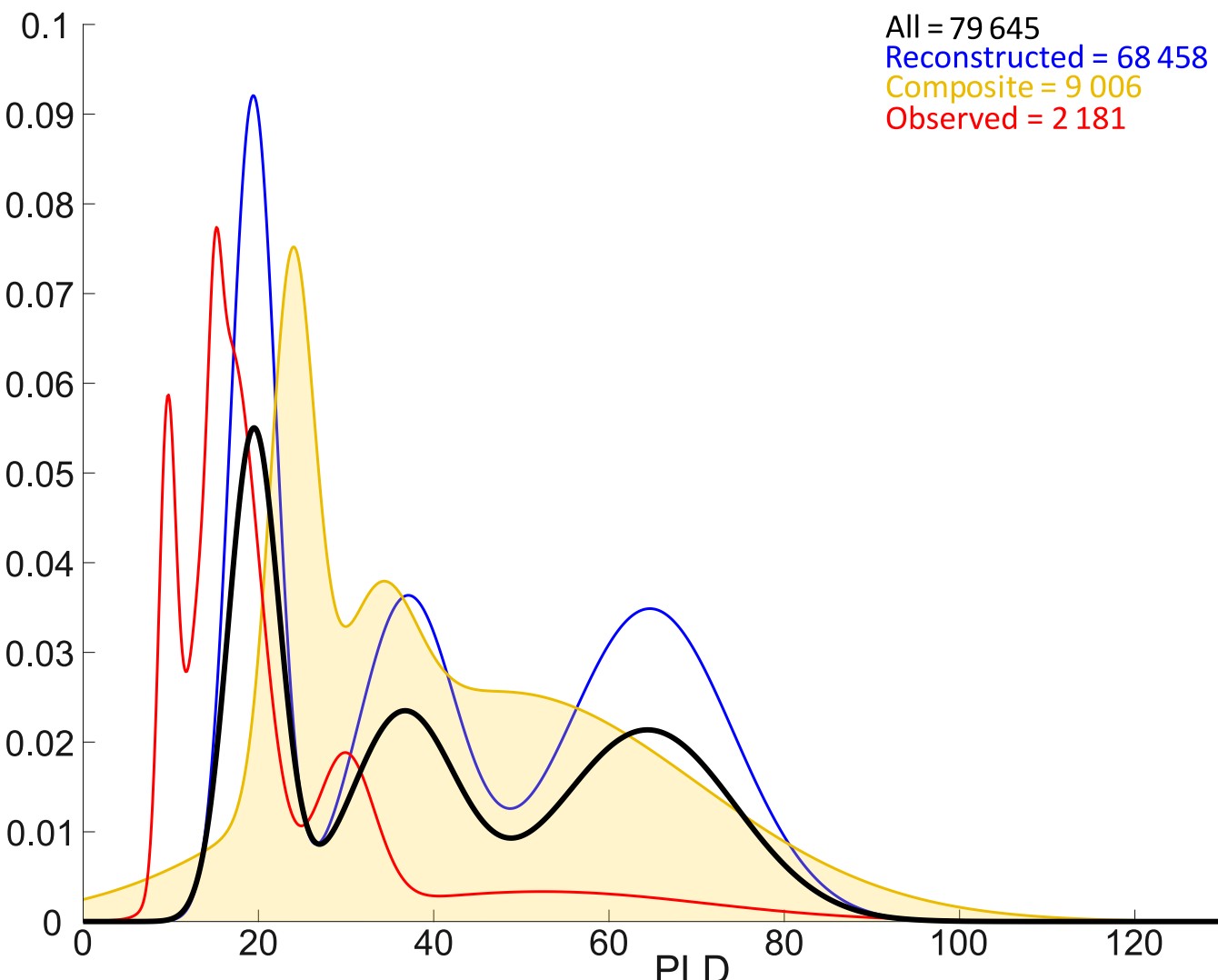

**Figure 7.** Density estimates (Gaussian mixture models) of all PLDs (bold black curve), observed PLDs (red curve), reconstructed PLDs (blue curve) and composite PLDs (estimated from observed settlement dates and reconstructed spawning dates, yellow curve). Note that while this figure represents the distribution of all PLDs across the database, shape of densities are also affected by non-uniform taxonomical sampling, as reported on Fig. 3.

settlement dates and reconstructed spawning dates) represent 9 006 entries (11.3% of total PLD data). In the database, observed PLDs range between 6.5 and 88 days and the distribution peaks around 10 days, 16 days and 29 days (Figure 7). Composite PLDs range from 3.9 to 250.5 days with peaks at 24 days, 34 days and 48 days. Reconstructed data span 6.2 to 191.8 days and the peaks around 20 days, 37 days and 65 days are very similar to those of all PLDs. PLDs of any category follow similar multi-modal distributions with three peaks (Figure 7). Peaks derived from all PLDs and those from reconstructed PLDs overlap,

whereas observed and composite data peaks tend to be shifted to lower values (except the first peak of composite data). It is worth noting that reconstructed data have a substantial influence on the final distribution of PLDs (Figure 7) as well as on the ones of spawning and settlement dates (Figure 6) since they are 5 times more numerous than observed data.

## 4 Discussion

### 4.1 Early-life traits analysis

Based on the current ecological literature, a large majority of Mediterranean coastal fish species are known to spawn (and settle) during a specific season or period, usually lasting two to four months (Tsikliras et al., 2010), which can be observed in the database with *Mullus barbatus* for instance (Suau & Vives, 1957). Contradicting this view of mostly-restricted phenology for coastal fish, the database highlights a more continuous phenomena with longer periods (spanning several months) during which spawning and settlement occur, both at the database scale (black bold curve in Figure 6) as well as at the species-level (e.g. *Diplodus sargus*). In fact, field studies are often limited in space and time, and/or in the variety of species and techniques (or may consider stocks only, such as Tsikliras et al. (2010)). They are rarely made simultaneously in different regions and often disregard inter-annual variability, even thought these scales are clearly relevant to best apprehend temperature-triggered spawning events (Di Stefano et al., 2022). In the future, more accurate information on early-life traits (spawning or settlement) could be obtained by carrying out regular sampling all over the year and within several regions.

Nonetheless, physiological studies such as those studying the gonadosomatic index, a good proxy of fish reproductive cycle (Conover, 1992; Tsikliras et al., 2010; Mouine et al., 2012), also support the idea that the species-specific periods favourable for spawning are longer than those derived locally. Our results thus advocate for compiling ~~long-term~~ extensive observations over large-scales before determining meaningful ecological envelopes. Note also that the gonadosomatic index evaluates adults' spawning potential, whereas here we assess spawning periods, including direct egg observations but most of all *a posteriori* information from individuals that have already survived the dispersal phase. As such, we are not estimating spawning dates from the entire population, but only from survivors, which could introduce a survivor bias. In fact, growth-selective mortality could play a role here so that traits of survivors from a given year favouring slow-growers may be different from those of another year favouring fast-growers (Pepin et al., 2014). Nevertheless, we here integrate information from different life stages (e.g. eggs, larvae, settlers and juveniles), suggesting they are affected by various selective pressures and mortality rates. We thus find that spawning process is longer than previously thought, so individuals sampled in a specific place could come from batch of eggs spawned from different periods, and not only from similar patches of peak spawning (Di Franco et al., 2015; Legrand et al., 2019).

In Figure 7, the distribution of all PLDs points out three categories: short (around 20 days, two to three weeks), medium (around 35 days, one month) and long (around 65 days, two months) PLDs. Same patterns appear for observed, composite and reconstructed PLDs, sometimes with little shifts. These three categories obtained with large statistics, including many observed otolithometry-derived short and medium PLDs, are consistent with previous studies (Raventos & Macpherson, 2001; Macpherson & Raventos, 2006) and could serve as a reference for future ecological or connectivity studies using generic

PLDs in the Mediterranean Sea, as done in Dubois et al. (2016) and Legrand et al. (2022). From a more evolutionary point of view, these dispersal timescales could be associated with typical residence times of coastal waters in order to balance auto-recruitment and larval export/import (Dubois et al., 2016). Residence times in the coastal ocean span 5-10 days in small embayments (Hernández-Carrasco et al., 2013), 2-20 days in large bays (Rubio et al., 2020) and range from 31 to 90 days in the entire Mediterranean according to a global modelling study (Liu et al., 2019). While residence times depend on local bathymetry (e.g. geometric enclosure) and (sub)mesoscale activity, these physical timescales are indeed in good agreement with biological timescales of dispersion revealed by our data compilation. It suggests a potential adaptation mechanism at the community level in temperate and open seascape, as previously documented in relatively-enclosed tropical reefs (Paris & Cowen, 2004). Studying Mediterranean fish species reared under controlled conditions, Lika et al. (2014) also suggest that dispersal is key to the coupling found between temperature and metabolic acceleration at larval stage.

Note also that early-life traits exhibit a certain plasticity linked to other abiotic factors. For *Diplodus* genus, Vigliola (1998) shows that short PLDs are often associated to spring and summer settlement (*Diplodus sargus*), whereas long PLDs are often seen during autumn and winter (*Diplodus vulgaris*), perhaps because larvae must endure coldest temperatures along their drift, limiting their growth (Green & Fisher, 2004). More recently, Raventos et al. (2021) document a tight link between PLD and environmental factors (seawater temperature in this case) prevailing during the pelagic dispersal.

It must be considered that early-life traits compiled here can be slightly biased by over-represented fish species. Observed PLDs are otolithometry data and these ones concern mainly Sparidae (*Diplodus sp.*, *Sarpa salpa*, *Oblada melanura*) but also *Chromis chromis*, *Coris julis* and *Symphodus sp.* genus, known for their short PLDs (Macpherson & Raventos, 2006). On the other hand, the long tails of distribution of all PLDs are due to the numerous samples of *Parablennius sp.* and *Lipophrys sp.*, two genus of Blenniidae for which males are taking care of eggs until hatching (Almada et al., 1992; Giacomello & Rasotto, 2005), as well as *Hippocampus sp.* exhibiting specific reproductive process (i.e. eggs and just-hatched larvae growing inside the male brood pouch and then drifting during eight weeks (Curtis & Vincent, 2006)). Indeed, the possible relationship linking long PLDs with fish species practicing egg parental care remains to be investigated, as this behaviour tends to protect the eggs from predation and favour the release of larger larvae, potentially limiting early-life mortality during pelagic dispersal.

## 4.2 Uncertainties resulting from sampling efforts

Sampling biases in such databases can arise due to spatially and/or temporally restricted sampling efforts, as well as to hetero-geneous taxonomic coverage. While former bias sources are usually linked to practical constraints, the latter may also reflect behavioural discrepancies. For instance, cryptobenthic species as Blenniidae or Gobiidae are rarely identified at species-level. In addition, they are difficult to observe in the field at all stages as they are hidden in seagrass meadows and macroalgae on rocky reefs, and are thus less studied, despite recognized anthropogenic impacts on these species (Brandl et al., 2018). In fact, investigations mainly focus on accessible habitats and assemblages, especially shallow coastal habitats identified as the depth range where most of coastal fishes are thought to settle (Sparidae, Labridae; Harmelin-Vivien et al., 1995; Cheminée et al., 2021). Nevertheless, settlement also occurs in deeper habitats of other types (e.g. sandy / muddy bottoms) for numerous taxa but with still few data up to now (except for well-studied species such as *Merluccius merluccius*; Druon et al., 2015).

The database covers almost three decades but with substantial gaps in the late 1990's/early 2000's, late 2000's and the most recent period (late 2010's/early 2020's). This is related to the lack of sampling effort during certain periods but also

to the evolution in sampling technique uses. In fact, highly destructive methods (such as dynamite and rotenone toxin) were essentially used to sample juveniles until the 1990's in parallel with classical methods such as nets and were progressively replaced by the non-extractive UVC method (Harmelin-Vivien et al., 1985) and light-traps. The low quantity of data sampled during the most recent period (around 2020s) is probably due to data that have not been processed or did not lead to a peer-reviewed publication yet, so that researchers are less willing to share their data. Note that the status of the ecosystems where

data were collected may have changed. In fact, as this database spans a long period and many places, anthropogenic pressures and climate change may have modified the habitat or fish populations' phenology, potentially leading to a shift in fish traits.

Future sampling efforts should aim at being more continuous over an annual cycle and cover all months / seasons as Díaz-Gil et al. (2019) in the Mediterranean Sea or Bograd et al. (2003) in the Californian Current System. In fact, our results show that spawning and settlement occur pretty much all year long when considering many fish species, which is not seen on the

345 distribution of sampling efforts. A noticeable difference is the clear prominence of sampling during summer as compared to winter (Figure 6). Same observations are made by Brady et al. (2020) in freshwater studies. Thus, harsh winter conditions may discourage researchers ~~to plan for~~ from planning fieldwork in coastal ~~waters, especially~~ areas, particularly for UVC involving divers. Note also the clear lack of sampling in August that may be linked to summer vacation periods of most European countries. Even though the combination of both direct and reconstructed data allows to partially overcome these sampling bias,

a temporally restricted sampling for settlement will undoubtedly reflect itself in the reconstructed spawning dates. For instance, as *Boops boops* settlement is only observed from April to June for rather short PLDs (around 15 days), spawning can only be ~~reconstructed in end of winter and spring~~ estimated as occurring at the end of winter and in the spring. In other words, it is not possible to verify for this species if the absence of spawning between July and December is due to biological reasons or due to an uneven sampling of the settlement. In fact, most published studies have been mainly focusing on shallow warm-season

organisms that are considered, as a potential common biased belief, as more ecologically- or fishery-relevant species than those with wide thermal tolerances and/or inhabiting greater depths, whereas they might be equally important.

Concerning space, the majority of projects constituting the database are especially targeting core or surroundings of marine reserves (a co-location procedure using Marine Protected Areas of National Statute from MedPAN and SPA/RAC (2021) reveals that it concerns more than 50% of entries; not shown), while the rest spreads out over various unprotected areas,

probably targeted by fisheries. The most represented reserves are Cabrera, Calanques and Port-Cros National Parks but also Cabo de Palos, Cerbère-Banyuls and North-Menorca marine reserves. Marine Protected Areas are known to house higher biomasses with larger fishes, i.e. more effective spawners (Lester et al., 2009; Edgar et al., 2014) and higher reproductive outputs (Marshall et al., 2019). Moreover, intense fishing pressure have been shown to shift reproductive and life history fish traits, which might ultimately impact the phenology of spawning and settlement (Heino et al., 2013). As such, this spatial

sampling bias may also slightly affect retrieved spawning and settlement periods.

The precision of both species identification and length determination is also dependent on the sampling methodology. For instance, UVC is a non-harvesting method so that the length measurement is only based on the visual appreciation of the

observer (to the nearest centimetre or half-centimetre, which can be a rough estimator for precise growth models), which have been shown to overestimate juvenile sizes (Edgar et al., 2004). Juvenile data may also be less accurate to determine early-life locations as active movements may affect connectivity (Di Franco et al., 2015). On the contrary, harvesting methods like bongo nets or traps consist in keeping organisms in solutions and then measure them precisely (to the nearest millimetre) in the laboratory. What can appear as small differences of sizes (one centimetre for instance) may indeed affect our results through the simplified categorization of life stages and reverberate on the precision of the reconstruction (e.g. the standard deviation of the trait). Note that the sensitivity of our results to size determination might differ from one species to another. For instance, it should be less important for fish species with specific body shapes, such as *Coris julis* which is a relatively long fish at adult stage, so that small imprecision would have little impact on age determination. No alternative predictor of age (such as fish weight) is used in the study as AmP growth curves are mainly based on length/age relationships and most studies record length only, especially when using no-take methods such as UVC (Harmelin-Vivien et al., 1985; Cheminée et al., 2017; Cuadros et al., 2019).

Since DEB theory conserves mass, future works may revisit the proposed reconstruction process by exploiting species-specific size/weight/shape relationships. This study exploits size information because of their wide availability, whereas weight measurements of fish early-life stages are more difficult to carry out in the field and depend on the sampling technique, explaining their scarcity.

A last source of uncertainties relies on the non-exhaustive compilation of data due to a lack of answers of the contacted researchers or an incomplete screening of literature. In fact, we focus on the entire Mediterranean Sea but we could only retrieve data from the North-Western Mediterranean basin and the Adriatic Sea. Limiting financial and human resources as well as complex geopolitical situation may explain why sampling is scarce, non-yet digitized or even absent, along some Mediterranean rim countries. Further studies could expand our methodology to integrate South-Western and Eastern Mediterranean basins.

### 4.3 Uncertainties resulting from the use of growth models derived from DEB theory

While the `std` model is well adapted to adults' life cycle, we retain here the `abj` model, best adapted to the entire life cycle as it explicitly models the fish early-life stages (when available in AmP website, see section 2.2). Different bioenergetics and environmental conditions indeed apply to sedentary and dispersive stages (transition from a passive pelagic life to an active coastal life) explaining the separation of the growth curve in two equations in the `abj` model (Kooijman et al., 2011; Kooijman, 2014). For remaining fish species, we have to exploit Von Bertalanffy growth curve of the `std` model. In fact, limited observational data implemented in the AmP dataset of these fish species might have prevented AmP contributors to define their `abj` parameters. Another possibility is that some fish species may have been well studied and are indeed known to follow the `std` life cycle. Nevertheless, the `abj` model is an one-parameter extension of the `std` model. When $L_j = L_b$, the former reduces to the latter and data can also be fitted with the `std` model, exhibiting similar or smaller mean relative error. Note that fisheries scientists often describe fish growth using the yield-per-recruit model of Beverton and Holt (1957) when performing stock assessment. By providing the expected life time yield per fish recruited into the stock at a specified age, it is used by fishery managers to understand and predict the biological and economical effects of fishing on the stocks and helps

them to take suitable measures to ensure sustainable exploitation. In this context, further work is needed to reach a consensus on a generalized growth model (encompassing both exponential larval growth and logistic adult growth) that could then be used in stock assessment exercises.

Development is highly sensitive to temperature, varying with location, season and depth (Dahlke et al., 2020). Moreover, the position of early-life stages in the water column is correlated to the size of the individual, also impacting food selection and thus changes in diet. However, due to the difficulty of getting local temperature and food data across the entire Mediterranean, unique $T$ and $f$ are uniformly drawn 50 times for each growth curve and the retained growth curve is an average of the former ones (see parameter range in Table 2). More realistic results could be obtained when considering evolving temperature and

food conditions encountered along the Lagrangian trajectory of a modelled larva, as done in Lagunes et al. (2024). Overall, the coupling of bioenergetics models with dispersal and ocean models holds a lot of promises, such as providing the mechanistic basis for projecting climate change effects on marine living resources (Rose et al., 2024).

Settlement and metamorphosis events are sometimes considered to occur concomitantly, especially in otolithometry studies (Di Franco & Guidetti, 2011), or settlement may also be considered as a part of metamorphosis, a behavioural event in an

415 extended-in-time morphological process (Vigliola & Harmelin-Vivien, 2001). Metamorphosis is however an ontogenic process (development biology) whereas settlement is rather an ecological event. In fact, metamorphosis is a more or less continuous process as the larvae acquire the ability to swim progressively toward the end of the dispersive phase. As metamorphosis is a fixed-in-time event in the DEB theory, we decide to separate metamorphosis and settlement in the mathematical reconstruction process. The randomly chosen lengths at settlement $L_{settl}$ are then considered different from lengths at metamorphosis $L_j$

determined by the DEB model, even if they can be close or equal. Moreover, previous tests fixing $L_{settl}$ equal to $L_j$ led to distorted growth curves, as this assumption is not applicable to all species. Finally, the length classification used for this study (based on literature) is a way to generalize models for Mediterranean coastal species, but classification limits are obviously species-specific and even individual-specific in reality.

For genus-based reconstruction, we choose to average all different growth curves of species from a same genus into a unique

growth curve for the genus (see Tables S5 and S6). This has not been done with families as there could be major differences among fish species belonging to the same family. Even if reconstructing data at genus level is questionable, this choice allows reconstructing more data and thus provide first insights on early-life traits of some less studied genus or species. Finally, this process points out a need to determine missing growth parameters in AmP in order to obtain more reliable growth models, missing for some highly exploited fish species such as *Mullus surmuletus*.

In this study, we combine field observations and DEB modelling, potentially resulting in two sources of errors. They are however independent since DEB parameters are initially estimated based on observational datasets that are not contained in our compilation. In fact, depending on the studied area, early-life stages could have faced particular environmental conditions, different predatory or competitive pressures, favouring for instance the emergence of various asymptotic lengths $L_i$ when modelling growth and explaining why we accounted for environmental variability within the DEB model. Note also that growth

models provided by AmP have a measure of goodness-of-fit through the mean relative error and symmetric mean squared error (Kooijman, 2022). Tests were conducted on one of the main species of the database (*Diplodus sargus*) to determine the impact

of this uncertainty on the reconstruction results (see Figure S4). Differences on reconstructed dates represent one to two days maximum. We disregard the intrinsic uncertainty of DEB growth models as we preferred to explicitly integrate the variability of environmental processes via the introduction of $f$ and $T$. Finally, as the DEB theory is based on metabolism and shape of the species (Kooijman, 2010), the impact of the data used to feed parameters of these growth models should be limited.

Note however that, even with well-fitted AmP growth models, our reconstructions might still be biased if their initial input datasets do not properly comprehend the full range of natural variability. In fact, many AmP parameter values are computed from data collected in the wild (e.g. for species of the *Diplodus* genus), but data are often limited in number, space and time. Also, some AmP parameters are determined based on laboratory experiments (i.e. under controlled abiotic conditions, as for *Atherina boyeri*) or estimated from other species of the same genus (e.g. the age at birth of *Diplodus sargus*, representing hatching, originate from *Diplodus puntazzo*'s one). Moreover, some species-specific models are built by considering observations pertaining to one given life stage. While the few models fitted on early-life data may distort adult part of the curve (e.g. *Diplodus puntazzo* or *Pagellus erythrinus*), most models originate from adults data only, suggesting less reliability of the early-life part of the growth curve, extrapolated from adults' one (e.g. *Chromis chromis* or *Epinephelus marginatus*). Others are fed with too few points (e.g. *Atherina boyeri* or *Uranoscopus scaber*). In this respect, all early-life entries compiled here can serve as a new independent dataset to improve some of AmP growth curves.

Finally, when comparing species-specific PLDs gathered in our database against those derived from otolithometry (known as a reliable estimator of PLDs) in literature, the range of values is generally consistent despite a few noticeable discrepancies. For instance, *Lithognathus mormyrus* and *Bothus podas* reconstructed PLDs are around 5 and 4 months respectively, much higher values than 1-month PLDs derived from otolithometry by Ayyildiz and Altin (2021) and by Macpherson and Raventos (2006) for each species respectively. While our estimates seem more aligned with typical PLDs of high-latitude slow-developing species adapted to cold climates, it is also possible that the small sample sizes for *Bothus podas* or the narrow sampling windows for *Lithognathus mormyrus* would by themselves explain the mismatches. In other words, a highly locally-adapted subpopulation surveyed by a too-restricted sampling could return PLDs that are not representative of the full variability of dispersal traits expected at basin-scale (impact of this sampling bias already observed in terrestrial studies; Wisz et al., 2008; Mentges et al., 2021). Another example of mismatch concerns *Chromis chromis*: the mean PLD of our database is about 3 times higher than PLDs documented in literature. To clarify reasons behind this mismatch, we compared reconstructed PLDs using (i) the AmP default fit and (ii) a fit using only 1/3 of smallest length observations (not shown). With this specific fit, reconstructed PLDs are more concordant with literature than those obtained with the default fit relying mainly on large lengths from adult observations. These two exemplary issues point out the constant necessity to test models against observations as well as to feed more high-quality observations of all stages into DEB growth models to better constrain parameters. Overall, it calls for more regular and tight interactions between field and theoretical ecologists.

## 4.4 Perspectives: future data reuse

First, this compilation of early-life dispersal traits (namely dates and locations of both spawning and settlement along with PLDs) for bipartite life cycle coastal fishes will allow, in combination with other trait databases (Teletchea & Teletchea, 2020),

to perform ecological analyses focusing on the phenotype variability for a given species or comparatively among various species at both organism and population levels. Previous research suggest a substantial sensitivity of fish early-life traits to ongoing climate change (Pankhurst & Munday, 2011; Llopiz et al., 2014; Donelson et al., 2019). This database covering almost three decades, which may be updated in near future as new observations become available, could foster further analyses of trait variability over longer, near climatic, temporal scales. When analysed in concert with environmental factors derived from satellites, operational ocean models or other aggregated datasets (e.g. Tyberghein et al., 2012; Assis et al., 2018), they would also allow studying traits' plasticity. Overall, it should contribute to reconciling ecological and evolutionary considerations. For instance, Harmelin-Vivien et al. (1995) state that *Diplodus sargus* and *Diplodus vulgaris* share same nurseries but at different seasons due to potential selective evolution. This database could help verifying if this kind of association occurs also in other environments and if it may evolve through time while climate changes.

Moreover, PLDs, spawning and settlement dates and locations should promote further connectivity studies and provide reliable constraints (by informing when and where dispersive stages start and end) for dispersal models at the population- or at the community-level with random samples drawn into different distribution functions, representative of Mediterranean bipartite coastal fish (such as general PLD values used in Dubois et al. (2016) or Legrand et al. (2022)). Of course, species-specific analyses of dispersal traits, such as PLD, can be easily carried out by filtering the database accordingly.

Connectivity and climatic studies could even be combined to better understand how marine assemblages are being re-shuffled spatially (e.g. moving poleward) while considering the plasticity of dispersal traits. Finally, the classical match/mismatch hypothesis (Cushing, 1990) could be explicitly tested based on modelling space-and-time-dependent spawning, dispersal and settlement together to better understand early-life mortality rates and how it controls interannual variations or long-term trends in fish recruitment (Hidalgo et al., 2019; Ferreira et al., 2023).

More generally, current eco-evolutionary theory postulates that traits' plasticity may arise from phenotypic and/or genetic components. Phenotypic plasticity describes the ability of an organism to change traits in response to environmental variations without genomic modifications. Although still being debated, genetic plasticity would refer to similar changes in a trait across multiple environments due to differences in allelic expression and to changing interactions among loci. Also called "genetic assimilation" and supposedly being transferable to next generations, it is when environmentally induced phenotypic variations become constitutively produced (i.e. being encoded in the genome, they no longer require environmental signals for expression). In both cases, the breadth of diversity encompassed by this database is proportional to (i) the number of species sampled, (ii) the absolute number of sampled individuals for a given species, (iii) the spatial representativeness of sampling across the entire distribution range of the given species, and (iv) the time coverage and temporal frequency of sampling over its existence (e.g. from speciation to extinction). As such, the reliability of the dispersal traits and of their variability contained within this database relates to the completeness of the input data (whose sampling gaps have been determined previously). In other words, the higher the number of future new entries, the more informative and accurate the database will become.

Following the application of FAIR principles (Findable Accessible Interoperable and Reusable studies and data; Wilkinson et al., 2016) in marine ecology, our original methodology could democratize the creation of large databases combining observed, modelled or mixed data from multiple sources and places. Considering the limited funding for research as well as elevated

costs and complications of field campaigns, we believe this kind of initiative could be further developed to limit data retention and data loss while increasing explanatory power of existing datasets, that may not be harmonized or accessible otherwise. As required by most public founders, instructions are to grant access to standardized, accessible, permanent and renewable records on historical data, generally stocked on paper or on obsolete storage technologies, otherwise destined to disappear. Data diffusion has become easier via the use of numerical sharing tools and online data repositories, favouring the development of similar initiatives in the Mediterranean Sea or elsewhere. We here present the first version of a database that, we hope, will be used and further developed by the research community. Indeed, the methodology has been fully described previously and the SEANOE data repository allows updates and expands of the database content. Such updates will concern (i) any newly observed entries by on-going research projects, (ii) ancient ones that authors could not share with us in due time, and (iii) the already reconstructed entries that could be recomputed as soon as the AmP database is being updated (both through better-fitted parameters and newly covered species). More generally, potential future works could aim at linking those databases automatically and relaying aggregated results in the well-known FishBase community-website (https://fishbase.se/).

*Acknowledgements.* Authors acknowledge support from Région Sud for M.DS's PhD grant and co-funding by Port-Cros National Park through a project supported by the Prince Albert II of Monaco Foundation. V.R. acknowledge partial support from the 4DMED-SEA project funded by ESA (contract No. 4000141547/23/IDT). Authors thank Maurice Libes for his help and advice on database formatting and on data repositories. Authors acknowledge Nathaniel Bensoussan for his valuable help in quickly handling the MAPAMED dataset and Witold Podlejski for his precious mathematical help and advice. For sharing data they collected, authors also thank the following data samplers and/or researchers part of the initial projects and studies: Franco Biagi, Laureline Chassaing, Bret Danilowicz, René Galzin, Antoni Garcia-Rubies, Adrien Goujard, Dulcic Jakov, Jean-Yves Jouvenel, Marine Leteurtrois, Nicolas Lucchini, Ciara O'Leary, Hector Torrado, Leonardo Tunesi.

*Code and data availability.* Data are freely available as a unique dataset and associated metadata on the data repository SEANOE at https://www.seanoe.org/data/00800/91148/ (Di Stefano et al., 2023). Matlab codes of DEB growth models are available on each species page of the Add My Pet website at https://www.bio.vu.nl/thb/deb/deblab/add_my_pet/ (in "Collection" and "Species Name").

*Author contributions.* Authors confirm contribution to the paper as follows:

– M.DS., D.N. and V.R. realized the study conception and design;

– V.R. and D.N. provided critical feedback on the experiments;

– M.DS. and D.N. performed the statistical models;

– M.DS. analyzed and interpreted the results, with contributions from V.R. and D.N.;

– M.DS. led in writing the manuscript, V.R. and D.N. reviewed the initial and final versions of the manuscript and provided critical feedback;

– I.A., G.A., P.A., G.B., A.B., D.B., A.Ca., I.C., C.C., A.Ch., R.C., A.Cu., A.DF., C.DG., T.E., R.F., FC.FH., JA.GC., P.G., L.G., JG.H., M.HV., M.H., H.H., JO.I., G.LM., L.LD., P.L., E.M., S.MS., M.Me., M.Mi., T.M., J.M., M.Mun., M.Mur., L.N., MP.O., J.P., A.PR., S.P., N.R., J.R., El.R., Er.R., S.R., A.S., T.T., D.Ve., L.V., D.Vr. provided data, reviewed the manuscript and provided critical feedback.

*Competing interests.* The authors declare that they have no conflict of interest.

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
