# Peer review of "Early-life dispersal traits of coastal fishes: an extensive database combining observations and growth models"

_Earth System Science Data, 2024_

## Author Comment (AC1)

- I very much agree with the authors about the ecological relevance of the early life stages of fish, and the need to become better organised on structuring information. The von Bertalanffy growth curve is indeed very popular, but most authors use the Beverton & Holt formulation that has the extra parameter t_0: the time before birth at which L=0. This obviously lacks any biological realism/relevance and results from fitting length-at-age data at 1, 2, ... years, excluding the early stages, which deviate for the von Bertalanffy growth curve. Only 231 of the 35600 fish species in fishbase have data on egg development, which illustrates the problem. The paper is well written and hopefully initiates an important data structuring effort.

  ⇨ Thank you very much for your positive appreciation and constructive comments. Following the first remarks from the reviewer, we added a few sentences in section 2.2.1 to better support our choice and in section 4.3 to put this study into a wider context (i.e. fishery science). Concerning the second point, as specified in the last paragraph of section 2.2, information at specific level was indeed missing for many fish species, explaining our choice for other classification levels (genus level reconstruction) when available.

- Development is very sensitive to local temperature, and temperature varies with location, season and, especially, depth of the fish. While the neonates typically live very close to the surface, later stages live in deeper (and cooler) waters, linked to their food. Food selection is very much coupled to size, so changes with growth. How is this considered?

  ⇨ We fully agree with this remark but this precise consideration would be overly complex and goes beyond the scope of our article. Unfortunately, local temperature is not considered here, which means depth, date and coordinates were not used to extract entry-specific temperature. Instead, for each growth curve, a temperature value is uniformly drawn from a range of typical temperature values observed in the Mediterranean Sea. Concerning food, the value is constant over the entire growth curve as we do not considered length in food selection in our study. The following paragraph has been added in section 4.3: "Development is highly sensitive to temperature, varying with location, season and depth (Dahlke et al, 2020). Moreover, the position of early-life stages in the water column is correlated to the size of the individual, also impacting food selection and thus changes in diet. However, due to the difficulty of getting local temperature and food data across the entire Mediterranean, unique T and f are uniformly drawn 50 times for each growth curve and the retained growth curve is an average of the former ones (see parameter range in Table 2). More realistic results could be obtained when considering evolving temperature and food conditions encountered along the Lagrangian trajectory of a modelled larva, as done in Lagunes et al, (2024). Overall, the coupling of bioenergetics models with dispersal and ocean models holds a lot of promises, such as providing the mechanistic basis for projecting climate change effects on marine living resources (Rose et al. 2024)".

- What are the plans to further develop and maintain the database? (see remark for lines 59, 76).

  ⇨ Please check the following comments.

- Remarks are made about further developments (line 365, section 4.4), but they do not make clear whether they will become part of the present database. What is the max capacity of Excel sheets? What about searching and extraction options? Who will do the maintenance? Is there a mechanism for submitting and curating new submissions? Are there any actions to place the database under the fishbase-umbrella?

  ⇨ The maximum capacity of Excel sheets is more than one million rows and 16 000 columns. Most of north-western Mediterranean coastal species being already included in the database, it is currently not necessary to consider a more complex structure (e.g. "Access") to manage it. Updating the database is possible though since the SEANOE data repository includes a process to revise, update and expand the content of the database. Our paper provides the methodology to do so with (i) any newly observed entries by on-going research projects, (ii) ancient ones that authors could not share with us in due time, and (iii) our reconstructed entries that could be revisited due to constant improvements of the parameters and of the species covered by the AddmyPet database (note that no specific procedure has been engaged due to a lack of funding and manpower). Concerning the integration of our results into *FishBase*, we are considering it for the future but no procedure has been engaged for now. We added a few sentences at the end of section 4.4 to reflect those statements.

Specific:

- 7 - dates -> data? Later formulations talk about missing information.
  ⇨ The word "data" is indeed more appropriate, thank you for your comment.

- 59, 76 - does "final database" implies that the database is what it is, and will not be maintained? No further additions? Or is meant "the database in its present state"?

  ⇨ "Final database" means in its present state and, in the context of the article, that the entire process of reconstruction has been applied to the database entries. The database is supposed to be updated, as highlighted above and now at the end of section 4.

- 85 - DEB theory takes "birth" as the event when feeding starts. Many newly hatched larvae have no mouth, or their mouth is still closed. It can take several days before the mouth becomes functional. The significance of this detail in a DEB context is in the investment into reproduction: the mother paid (via the yolk sac) for all development until the start of feeding. Weight at birth is in the AmP collection derived from the (typical) volume of an egg at spawning, assuming a specific density of 1 g/cm^3. The values on egg development in fishbase typically refer to hatch, not to birth.

  ⇨ The expression "just-hatched" comes from Catalan et al. (2014), as are the size ranges used to categorize early-life stages, so we consistently keep this wording throughout the entire manuscript. However, we fully agree with your remark and the following paragraph has been added in section 2.2.2: "Note that, strictly speaking,

hatch differs from birth. Hatch is the event when larvae free themselves from the egg membrane; birth generally refers to the time when larvae start feeding, as in some cases mouth needs slightly more time to open. During this short time interval (spanning a few hours to a few days for slow-development species), development relies on the yolk sac, whose energy is a function of the mother's investment into reproduction (Kooijman, 2010). Here, we disregard the distinction between hatch and birth and the associated duration by assuming that hatching is the beginning of fish early-life stages".

- 107 - The use of length for the early life stages is a bit tricky since they can change in shape substantially. DEB theory conserves mass, making mass more valuable than length.

  ⇨ We agree with this assertion. However, we had no choice but to proceed with available data, as most of them are based on length only. Depending on the sampling method, it is indeed easier to precisely measure length than weight. We add the following sentence in section 4.2: "Since DEB theory conserves mass, future works may revisit the proposed reconstruction process by exploiting species-specific size/weight/shape relationships. This study exploits size information because of their wide availability, whereas weight measurements of fish early-life stages are more difficult to carry out in the field and depend on the sampling technique, explaining their scarcity."

- 130 - Although the std DEB model simplifies to a von Bertalanffy curve AT CONSTANT FOOD AND TEMPERATURE for length-at-age, it differs from it at varying food. Many AmP fish-entries have varying food. The von Bertalanffy growth rate is not a DEB parameter (but can be derived from DEB parameters in combination with food and temperature). DEB theory also deals with reproduction, while von Bertalanffy (or better Pütter) does not. This can be used to understand that fecundity is approximately proportional to weight and increases non-linearly with length.

  ⇨ Std model indeed differs from von Bertalanffy growth curve when food is varying. However, as explained above, food and temperature conditions have been randomly drawn from temperature and food typical of the Mediterranean Sea for each growth curve, and all averaged to get the final growth curve. The sentences added in section 4.3 to address a previous comment (see above) also clarifies this point. The links between reproduction, maternal conditions and early-life stages are out of the scope of the present study.

- Table 1 - $t\_j$ is not a parameter, but depends on DEB parameters and food availability, such as the growth rate at the end of the exponential stage equals that at the start of the von Bertalanffy stage. So, it is not possible to see the transition in a time-length curve as is clear from Figure 2.

  ⇨ Thanks for this remark. However, for the sake of legibility, we kept the word "parameter" and added explanations about dependence to food and temperature (see the caption of Table 1.) Concerning Figure 2, we fully agree with your remark as this figure is only schematic and not to scale (see changes in the caption of Figure 2).

- Figure 2 - In DEB theory, length-at-time is more complex during the embryo-stage due to the depletion of reserve. But since L_b is typically small for ray-finned fish, this might be a detail in the present context.

  ⇨ In fact, that is the case for most of studied species, except for some specific genus such as *Atherina sp.* where L_b has been purposely shifted to higher values. This remark is stated in the third paragraph of section 2.2.2.

- 159 - This only applies for a given food type. Since length increase during ontogeny is enormous in ray-finned fish, size-dependent changes in diet are the rule, rather than the exception, and f is no longer restricted to the interval (0,1).

  ⇨ Thanks for bringing this up. We think that it does not apply to our work that is exclusively focused on early-life stages, which are characterized by similar diet (except concerning the subtle difference between internal yolk-sac and external feeding, previously acknowledged).

- Figure 3 - I could not find the family Tunidae in fishbase or Catalog of life. Do you mean Scombridae?

  ⇨ Thank you for pointing it out. It indeed represents the Scombridae family, but it was labelled as Tunidae in the collected data. We harmonized taxa as best as we could (see section 1 of the supplementary material), but some taxonomical inconsistencies may remain. The Scombridae family being disregarded from our analyses (they are not considered coastal species, as explained in the section 1 of the supplementary material), these entries have thus been removed from the database and the figure.

- 295 - https://www.zotero.org/groups/500643/deb_library/items/RUCFIFB3/item-list Lika et al 2014 found a coupling between temperature (summer vs winter spawners) and the acceleration factor in Mediterranean Perciformes. The paper suggests that dispersal is key to this.

  ⇨ Thank you for this valuable reference. The following sentence has been added in section 4.2: "Studying Mediterranean fish species reared under controlled conditions, Lika et al (2014) also suggest that dispersal is key to the coupling between temperature and metabolic acceleration at larval stage.".

- 374 - The abj model is an one-parameter extension of the std model. If the maturity-parameter E_Hj is close to E_Hb, the length at the end of acceleration L_j will be close to that at birth L_b. Indeed, if E_Hj=E_Hb, we have L_j=L_b and the abj model reduces to the std model. The choice for std or abj is typically made at the family or higher levels, not at the species level. All entries for which a std model is fitted, can also be fitted with an abj model, with the same or smaller mean relative error.

  ⇨ Thank you for your comment. The following paragraph has been added to section 4.3: "Nevertheless, the \texttt{abj} model is an one-parameter extension of the \texttt{std} model. When $L_j$=$L_b$, the former reduces to the latter and data can also be fitted with the \texttt{std} model exhibiting similar or smaller mean relative error."

- 408 - I fully agree with the remark that the AmP collection has shortcomings and see the database as part of a long-term maturation process, where entries with little or unreliable data are replaced and updated. The hope is that researchers will see the benefits of a high-quality database, recognize what info is essential, and start the collect data with DEB theory in mind.

  ⇨ Thank you very much for your comment.

- 429 - I very much agree with this remark.

  ⇨ Thank you very much.
* * *
**Referee 2:**

Manuscript:

- I'm not sure the usage of long-term in the title is fitting. While the collective database of years spans 29 years, most of the data may not be from long-term data sets. I would prefer the wording to be labelled as something such as "extensive" or "comprehensive" database, with the former being how the authors termed the collection themselves on line 47 as extensive compilation.

  ⇨ The word "extensive" is kept. See changes in title.

- Line 93, change "thanks to" to "using a", same line might also want to explain that the details of this method will be described in detail in the following section 2.2.

  ⇨ Thank you for your comment, changes have been made accordingly.

- Line 140, check for consistency in wording across documents and the database, the supplement uses eggs and small larvae, while this line uses eggs and just-hatched larvae. It would also be helpful if the authors addressed the caveats of using this classification, for example, do they have an opinion on whether most of the species included in this database generally fit this classification model that was meant as a general concept? This was addressed some on the last sentence on line 163-164, but I don't know if it is also marked as such in the database notes or not, which would be helpful.

  ⇨ Thank you for your comment, the wording has been modified for coherence across the manuscript, the supplementary material and the database. As in the first paragraph of section 2.1.2, the first paragraph of section 2.2 and in a previous response to another referee (see above), we adopted this classification based on published literature. It also sticks to the DEB model classification, supported by its own literature. As many classifications, wording and definitions exist, we tried to be as accurate as possible. For species whose development would be slightly different, we adapted the model to consider their specificities (section 2.1.2). For clarity,

references and length classes choices have been added to the metadata of the database in SEANOE.

- Section 3.1, it would be helpful to also provide summary information on depth, and I also wonder if Fig 4 presents the data in the best way, I might have a look at displaying each location with an equal sized dot so we can get a better feel for the regional comprehensiveness and colour the circles based on the number of entries sampled per node, that way some of the crowding won't overshadow understanding of geographic coverage. Additionally, we don't have a good feel for how many species are represented in the samples geographically, which would be useful to have visualized.

  ⇨ Thank you for your comment, the following paragraph has been added in section 3.1: "They have been generally collected near the surface: 90\% of the data are associated to depths ranging from 0 to 10 meters, while the maximum depth recorded in the database is 175 m, sampled with a plankton net". Moreover, thank you for the pertinent suggestion, as Figure 4 has been redrawn including coloured points.

- In general, some of the presentation choices present a challenge, especially because the 110000 entries are all treated the same despite knowing these are giving more weight to 15 or so families that represent the bulk of the entries. I find this particularly troublesome when looking at a figure like #7, because I can't put much weight into understanding how many of the families, species, etc. I am looking at in that database and how they end up weighting the observations graphed.

  ⇨ In section 3.2, we analysed the distribution of dispersal traits of the entire database across all taxa. We also discussed about the issues of having an unbalanced representation of species in the last paragraph of section 4.1. Following your suggestion, we have added a comment in caption of Figure 7 to avoid future misinterpretation. Note also that future readers can investigate species-specific PLD by filtering the database accordingly (a sentence has been added in section 4.4 to clarify this point).

- Lastly, over 80% of the dates are "reconstructed" so it would be helpful if the paper was clearer on best practices for reuse of this information and caveats, as we already see from Fig. 6 and 7 (which was helpful to include) that they may underestimate seasonality and/or overestimate PLD (although again without knowing the family/species details it is difficult to say this outright is really a pattern or not as species-specific details of which is observed versus estimated will critically matter) as pointed out in the earlier comment. I'm just left with wanting more context to be provided in section 4.3 for people that might want to reuse the database information have some clear recommendations. I appreciate the frankness about when there have been mismatches, as presented for example in the last paragraph of section 4.3, but that leads me asking, what is reliable, what isn't, what are the bounds of including a sensitivity analysis if including the estimates in a model for example.

  ⇨ We are perfectly aware of these caveats but they are unavoidable since our aim is to study dispersal traits across taxa from a compilation of numerous (yet, biased) available observations. To reinforce this statement, we add the following sentence in section 4.4: "More generally, current eco-evolutionary theory postulates that traits' plasticity may arise from phenotypic and/or genetic components. Phenotypic plasticity describes the ability of an organism to change traits in response to environmental variations without genomic modifications. Although still being debated, genetic plasticity would refer to similar changes in a trait across multiple environments due to differences in allelic expression and to changing interactions

among loci. Also called "genetic assimilation" and supposedly being transferable to next generations, it is when environmentally induced phenotypic variations become constitutively produced (i.e. being encoded in the genome, they no longer require environmental signals for expression). In both cases, the breadth of diversity encompassed by this database is proportional to (i) the number of species sampled, (ii) the absolute number of sampled individuals for a given species, (iii) the spatial representativeness of sampling across the entire distribution range of the given species, and (iv) the time coverage and temporal frequency of sampling over its existence (e.g. from speciation to extinction). As such, the reliability of the dispersal traits and of their variability contained within this database relates to the completeness of the input data (whose sampling gaps have been determined previously). In other words, the higher the number of future new entries, the more informative and accurate the database will become".

CSV file:

- The special characters did not display well in the CSV file nor import well into Excel, all of these should be double checked please. For example, one series of data had the following data for a site: Marseille - Endoume - Petits Fonds H√©t√©rog√®nes. Another site was shown as J√°vea.
  ⇨ We wanted to keep the name of the site in the initial language, but the special character issue did not happen during our tests so thank you for pointing it out, it has thus been changed accordingly.

- The CSV file was missing DOI direct links for each of the reference datasets (called oddly Projects when I would prefer it to be called reference or source), DOIs to the original data are essential to have, otherwise it takes multiple steps for someone using the database to get back to the original datasets to read more comprehensive information about the data because it would require piecing together information from the references which can be tedious.
  ⇨ We add a column called "DOI". We kept the Project column as some entries do not have DOI labelled.

General considerations:

- One is left mostly contemplating what as well the future of this dataset will be after reading section 4.4. As is, this mostly seems to be a meta-analysis of 44 studies which one would anticipate the authors themselves may reuse for different analyses rather than what end-users would consider a "long-term database". One is left wondering about the upkeep and broader dissemination. For example, is this information going to be added for each species included to fishbase?
  ⇨ The database will obviously be used as a material in other projects of our own, but we sincerely hope that it will be used by other researchers as a basis for their projects. Updating the database would be possible as the SEANOE data repository includes this system. However, in our case, it would only concern errors or improvements of data for species for which AddmyPet information is later available. Concerning the integration of our results into fishbase, we are currently considering it, but no procedure has been engaged for now.

- The dataset brings together 44 datasets and provides some new estimates, but perhaps its biggest value is the authors' evaluation on noting significant gaps in geography, species, and temporal coverage despite having almost 30 years of information. Specific comments about these gaps would be beneficial to include clearly

in the abstract and conclusions as a message / recommendation to the monitoring and research community on improving comprehensiveness going forward.
⇨ Thank you for pointing it out. We absolutely agree and tried to highlight this information in the end of the abstract.
* * *
**Referee 3:**

General comments:

- The manuscript lead by Di Stefano, Nerini, Rossi and a suite of collaborators describes and reports on development of a database of early-life dispersal traits of coastal fishes based on data from the western Mediterranean. The database consists of 111866 entries merged from 44 datasets from sample collections from 1993 to 2021. An extensive set of metadata serve to characterize each entry, which generally consists of the information from a single individual (92823) or several specimens. The database includes information about spawning coordinates (17130) or settlement coordinates (101536). The database includes the reconstruction of missing data using dynamic energy budgets (DEB) models.

  Overall, this unique database represents a potentially very valuable asset for scientists interested in the study of early life history stages. The collation of data from a variety of studies in a harmonized manner provides an archive of information that would likely not be available because data from each investigation could be lost in filling cabinets, dead computer drives, or scattered among various archives that would have to be harmonized for each investigator interested in the material. The use of DEB models to infer spawning date, pelagic larval duration, or settlement dates represents a standardized product that adds value to each observation. As a result, anyone having concerns about the information contained in the database can revisit the approach taken to determine how the overall results could be altered.

  The manuscript itself details the approach to harmonizing the information, including an assessment of the collection methods, that provides a comprehensive and repeatable description of the process followed by the contributors. The writing is somewhat uneven, with more grammatical issues (e.g., consistent use of verb tense, direct translation of some terms) from the introduction to the results about the nature of the database, while the discussion was rather more consistent in terms of the quality of the writing. The discussion addresses important sources of uncertainty in the database itself and the elements, but one concern not discussed is that collections over a 29-year period, from a variety of locations, may have been affected by the changing status of the ecosystems in which data were collected. The influence of anthropogenic pressures and climate change could be something to evaluate in a more profound assessment of the data, but there should be some discussion about the potential consequences of changing baselines. Dealing with the issue per se is well beyond the scope of the harmonization process but it is seldom considered and how that could affect the data collated from multiple sources. This likely represents a minor addition but it should be acknowledged as clearly as the other elements of uncertainty.

  My overall assessment is that this work represents a valuable contribution to the scientific community. The data reported in the dataset was developed based on a consistent and repeatable approach. Note that the data are delimited using semicolons, not commas.

⇨ Thank you very much for your valuable comments. The following paragraph has been added in section 4.2: "Note that the status of the ecosystems where data were collected may have changed. In fact, as this database spans a long period and many places, anthropogenic pressures and climate change may have changed the habitat or fish populations phenology, potentially leading to a shift in fish traits.".

Specific comments:

- L.36 "…highly studied over several decades with…"
  L.50 "…we aim to gather…"
  L.52 "…one original aspect of this work…2020) which allows us to enrich observations…"
  L.54 "…to provide evidence and interpret…"
  L.56 "…data collected over several decades with…"
  L.61 "…allow us to describe overall taxonomic and spatiotemporal coverage, and evaluate potential…"
  L.71 "Data were collected along…"
  L.74 The authors need to define UVC at the first occurrence in the text. Currently, the definition appears at line 200, which caused confusion as I worked my way through the document.
  L.111 "One unique feature of this database is the compilation…"
  L.123 "…data, which validates and substantiates our choice…"
  L.126 "It helps to provide a realistic physiological basis of growth dynamics…"
  L.160 "…0.7, 1] was selected arbitrarily, because the individuals collected have…"
  L.170 Spell out fifty-five: when starting a sentence with a number always use words; same applies when starting with a Latin species name.
  L.328 "…researchers from planning fieldwork in coastal areas, particularly for UVC…"
  L.333 "…15 days), spawning can only be estimated as occurring at the end of winter and in the spring."
  ⇨ All changes have been considered, thank you for your thoroughness.

- L.130 The variables describing different aspects of body length should be described in the body of the text, not in the legend (although they will need to be repeated) to Table 1.
  ⇨ Parameter description has been added in table 1 straightforwardly.

- L.150-155 It is important that the database stands alone, so a summary of the parameters applied in the DEB models should be summarized in the supplemental material, and linked to each of the 44 datasets, so that the work can be reproduced independent of the AmP website.

  ⇨ The equations used are equivalent for all datasets (basis and exceptions are described in section 2.2). They depend on the DEB parameters that have been extracted from the AmP database on 11 July 2022, reported in supplementary material in section 2. This information has also been added at the end of section 2.2.2. Note that the content of the AmP database is reliable but dynamic: it is supposed to be updated by the work of other researchers at any time. The only way to consider this feature would be to automatically link and update each database online, which is clearly out-of-scope of the present work. Note also that to address a similar comment made by reviewer 1, we added a paragraph at the end of section 4.4 to guide future developments by the community.

- L.154 Lsettl varies among individuals. How is this dealt with in the harmonization of the datasets?

⇨ The following paragraph precises that, for each entry of the database, Lsettl has been randomly drawn in a range between 0.6 and 1.2 cm. It is based on our initial categorisation of stages, with a few exceptions also described in section 2.1.2.

- L.157 I don't understand what is meant by "ranging in reasonable intervals"?

  ⇨ "Reasonable" has been changed to "realistic" in this section. All intervals are justified in the following sentences of the paragraph (Mediterranean temperatures, favourable food range, observed settlement lengths).